# Simultaneous multi-signal quantification for highly precise serodiagnosis utilizing a rationally constructed platform

Yuxin Liu [1,2], Zheng Wei[1,2], Jing Zhou [1]*, & Zhanfang Ma[1]*

Serodiagnosis with a single quantification method suffers from high false positive/negative rates. In this study, a three-channel platform with an accessional instrumented system was constructed for simultaneous electrochemical, luminescent, and photothermal quantification of $H_2S$, a bio-indicator for acute pancreatitis (AP) diagnosis. Utilizing the specific reaction between platform and $H_2S$, the three-channel platform showed high sensitivity and selectivity in the biological $H_2S$ concentration range. The three-channel platform was also feasible for identifying the difference in the plasma $H_2S$ concentrations of AP and normal mice. More importantly, the precision of AP serodiagnosis was significantly improved (>99.0%) using the three-signal method based on the three-channel platform and an optimized threshold, which was clearly higher than that of the single- or two-signal methods (79.5%–94.1%). Our study highlights the importance of constructing a multichannel platform for the simultaneous multi-signal quantification of bio-indicators, and provides rigorous ways to improve the precision of medical serodiagnosis.

[1] Department of Chemistry & Beijing Key Laboratory for Optical Materials and Photonic Devices, Capital Normal University, Beijing 100048, China. [2] These authors contributed equally: Yuxin Liu, Zheng Wei. *email: jingzhou@cnu.edu.cn; mazhanfang@cnu.edu.cn

Hydrogen sulfide (H$_2$S) is an essential gaseous molecule in various biological processes, conducting signals, and inducing responses between cells and tissues[1,2]. An elevated plasma concentration of H$_2$S is also recently shown to be associated with acute pancreatitis (AP), a critical disease with high early mortality rates[3]. Therefore, the evaluation of plasma H$_2$S concentrations may be useful in the serodiagnosis of AP. A variety of single-signal methods for AP serodiagnosis have recently been demonstrated, which are performed efficiently in AP[4–6]. However, it has been noted that, when only one signal is used to evaluate the plasma H$_2$S concentration, it is strongly influenced by the biological environment and operating conditions, which may result in elevated false-positive or -negative results and reduce the precision of the serodiagnosis.

The development of a simultaneous multi-signal method to quantify bio-indicators should counter the disadvantages of single-signal methods and provide much more information on the bio-indicator concentration in a single process[7–9]. This will significantly reduce the false-positive and negative rates in medical serodiagnostics. Therefore, the rational construction of an efficient multichannel platform by combining applicable quantification method is vital. Typical quantification methods are based on electrochemical, optical, and thermal parameters[10–13]. Whereas electrochemical methods have ultrahigh sensitivity and excellent reproducibility, optical and thermal methods are competitive in the rapid and accurate quantification of bio-indicators, so all three are extremely important in various serodiagnostic applications[14–19]. Therefore, the development of a multifunctional probe that combines the above quantification of a bio-indicator is a rational starting point in constructing a multichannel platform. However, to ensure that the sensitivity of electrochemical quantification is maintained, the multifunctional probe must be used to modify the surface of the electrode to improve the electron transport efficiency and mass transfer rate[20,21]. Moreover, unlike the optical signal, the thermal signal of the probe cannot be captured during the electrochemical quantification process because traditional temperature-measuring equipment cannot be used to directly and precisely monitor the microscopic temperature changes of the probe on the electrode surface. Therefore, determining the temperature of the probe on the electrode in situ becomes a major strategic issue in constructing a three-channel platform for bio-quantification.

Fortunately, recent advances have produced probes with heat-sensitive luminescence that are efficient nanothermometers for monitoring temperature on the microscale[22–26]. Therefore, microscopic temperature changes can be read by measuring the luminescence intensity, which provides an applicable strategy for determining the temperature of the probe on the electrode in situ[27]. Among all kinds of luminescent nanothermometers available, rare-earth-based nanothermometers have adjustable emissions under near-infrared excitation, with high signal-to-noise ratio and excellent stability[28–41]. More importantly, by doping the probe with various rare-earth ions and generating a core-shell structure, the rare-earth-based nanothermometer can emit orthogonal multi-peak luminescence for the low-interference and simultaneous determination of temperature and quantification of the bio-indicator[42–47]. As a result of the above theoretical basis, a multichannel platform could be established for simultaneously multi-signal quantification.

Here, by combining the advantages of electrochemical, luminescent, and photothermal methods, we rationally constructed a three-channel platform for the simultaneous three-signal quantification of H$_2$S by modifying a glassy carbon electrode with a luminescent rare-earth-based nanothermometer NaYbF$_4$:Er@NaLuF$_4$ (RENPs) and Cu-alginate (Cu–ALG) gel. The performance of this three-channel platform in quantifying H$_2$S was studied in vitro and in vivo using an established AP mouse model. The improved precision of AP serodiagnosis with the three-channel platform relative to traditional single- and two-signal methods was also evaluated in detail.

## Results

**Rationally construction of three-channel platform.** The three-channel H$_2$S quantification platform with accessional instrumented system was rationally designed and constructed as shown in Fig. 1a. Uniform RENPs with high dispersity and a narrow size distribution ($19.36 \pm 1.69$ nm) were prepared with a typical solvothermal method (Fig. 1b; Supplementary Fig. 1A, B). High-resolution transmission electron microscopy and their powder X-ray diffraction patterns demonstrated that the RENPs had high hexagonal crystallinity, with an observable (001) lattice plane (Supplementary Fig. 1C, D). The energy dispersive X-ray surface scanning results also showed that the NaLuF$_4$ homogeneously coated NaYbF$_4$:Er, which confirmed the successful preparation of the RENPs with a core-shell structure (Fig. 1c). Under 980-nm laser irradiation, the as-prepared RENPs had strong upconversion luminescence (UCL) emission peaks centred at 525 nm ($^2H_{11/2} \to {}^4I_{15/2}$), 545 nm ($^2S_{3/2} \to {}^4I_{15/2}$), and 654 nm ($^4F_{9/2} \to {}^4I_{15/2}$), and a short-wave infrared (SWIR) emission peak centred at 1550 nm ($^4I_{13/2} \to {}^4I_{15/2}$) (Supplementary Fig. 2). It was noteworthy that all luminescence intensities were significantly enhanced after NaLuF$_4$ inner shell coating, contributing to the inhibition of multi-phonon relaxation between the emitter and the adsorbed ligands[48]. In addition, due to the thermal-induced re-distribution in the populations of energy levels, the UCLs at 525 nm and 545 nm were thermo-sensitive, whose natural logarithm value of ratio (Ln [$I_{525}/I_{545}$]) was linearly related with inverse temperature in Kelvin unit ($Y = 1.9672–0.6411 \times$, $R = 0.9969$; Fig. 1d; Supplementary Fig. 3)[49–51]. Estimating from the UCL signal changes in response to temperature changes, RENPs were good luminescent nanothermometers with a relatively high sensitivity of 0.0108 K$^{-1}$ and low temperature uncertainty of 0.2559 K ~273–353 K (Supplementary Fig. 4), which was comparable with most previously reported nanothermometers, and was feasible to quantify temperature accurately (Supplementary Table 1). Though some of previously reported molecule thermometers also showed advanced temperature resolution and sensitivity[52–54], they mainly relied on single-band emission without an internal reference, which may be affected by complex biological samples. Therefore, with adequate ratiometric luminescent thermometry performance, it is reasonable to use RENPs for sensitive and accurate temperature sensing. With multiple emissions in orthogonal spectral range and advanced thermal-responsive capacities, RENPs were rational nanophosphor for simultaneous and independent luminescent and temperature sensing.

Based on the as-prepared RENPs, the three-channel platform was constructed by modifying clean glassy carbon electrode with RENP-contained copper-alginate gel (RENP–Cu–ALG). Scanning electronic microscopic images and energy dispersive X-ray surface scanning results suggested that the RENP–Cu–ALG gel was well distributed on the electrode (Fig. 1e, f). Element quantification results demonstrated that the optimized amount of RENPs on each electrode is $1.073 \pm 0.195$ mg (Supplementary Fig. 5). The RENP–Cu–ALG-modified electrode had a high signal current peak and strong luminescence in both the visible and SWIR spectral ranges, which contributed to the electronic signal of Cu–ALG and the luminescence of the RENPs (Fig. 1g–j). These results demonstrated that the RENP–Cu–ALG used to modify the electrode combined both outstanding electrochemical and luminescence capacities.

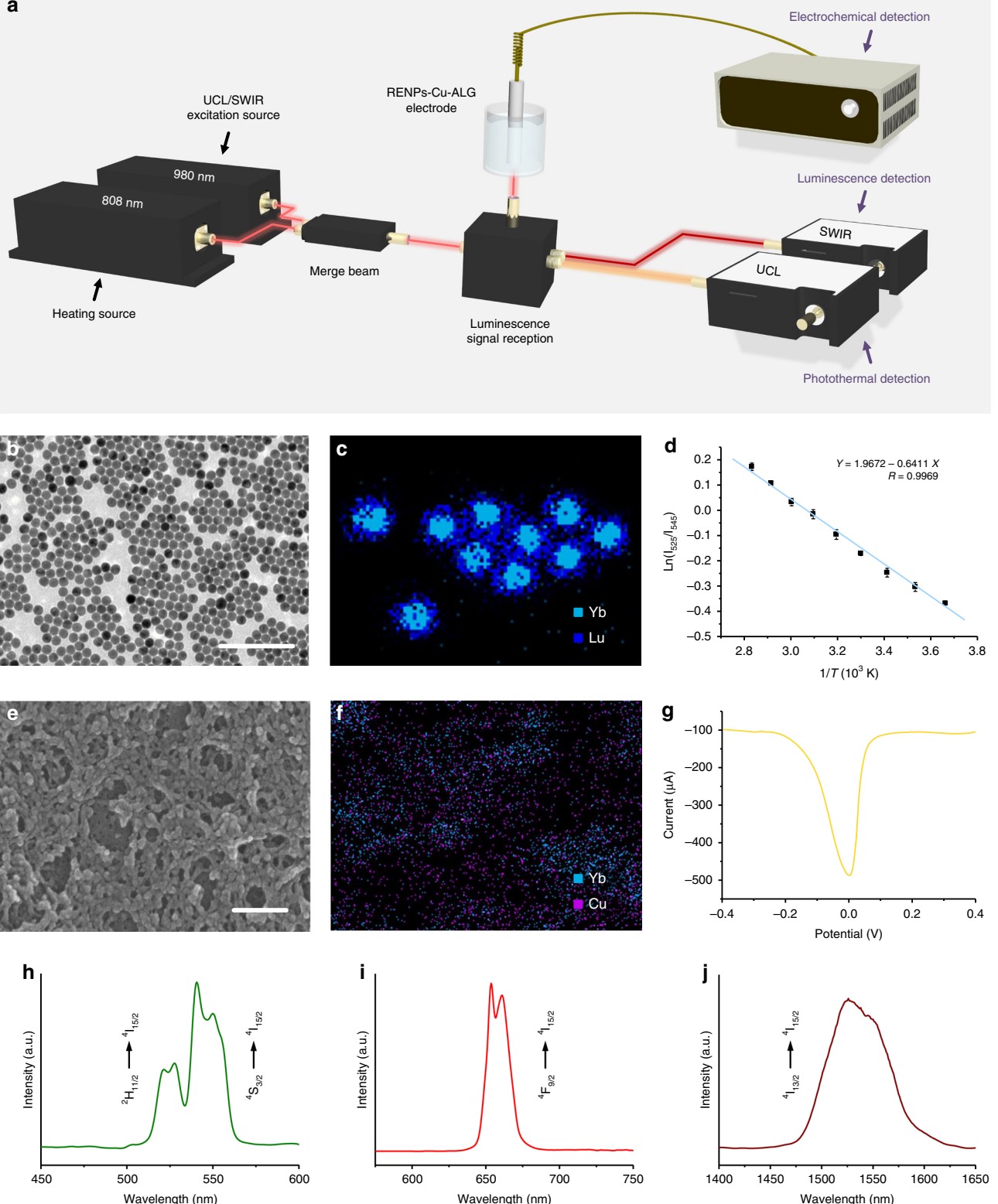

**Fig. 1** Characterization of three-channel platform. **a** Schematic presentation of as-designed three-channel platform based on RENPs–Cu-ALG-gel-modified electrode. Transmission electron microscope image (**b**) and energy dispersive X-ray surface scan (**c**) of RENPs. **d** A plot of $Ln(I_{525}/I_{545})$ versus $1/T$ to calibrate the thermometric scale for RENPs. Scanning electron microscope image (**e**) and energy dispersive X-ray surface scan (**f**) of three-channel platform. SWV curve (**g**), green UCL (**h**), red UCL (**i**), and SWIR luminescence (**j**) spectrum of three-channel platform. Scale bar, 200 nm. Data are represented as mean ± SD ($n = 3$).

**Simultaneous H$_2$S quantification by three-signal channel**. After confirming successful construction, the determination mechanism of platform was then studied. In the presence of H$_2$S, insoluble semiconducting CuS formed on the surface of the electrode in situ (Supplementary Fig. 6 and Eqs. (1–3)). Electrochemical impedance spectroscopy showed that the resistance of the RENP–Cu–ALG gel was significantly higher in the presence of H$_2$S than in its absence, so that the current signal decreased as the H$_2$S concentration (C$_{H2S}$) increased

(Fig. 2a, b). The formed CuS also had strong absorbance in a wide spectral region (600–785 nm), but relatively low absorbance in the SWIR spectral range (1400–1650 nm; Fig. 2c). Therefore, the red luminescence of the RENPs centred at 654 nm was selectively quenched by the inner filter effect of the CuS that formed in the presence of H$_2$S, whereas the SWIR luminescence was nonresponsive to H$_2$S and could be used as a reference for the quantification of the ratiometric luminescence with reduced environmental noise.

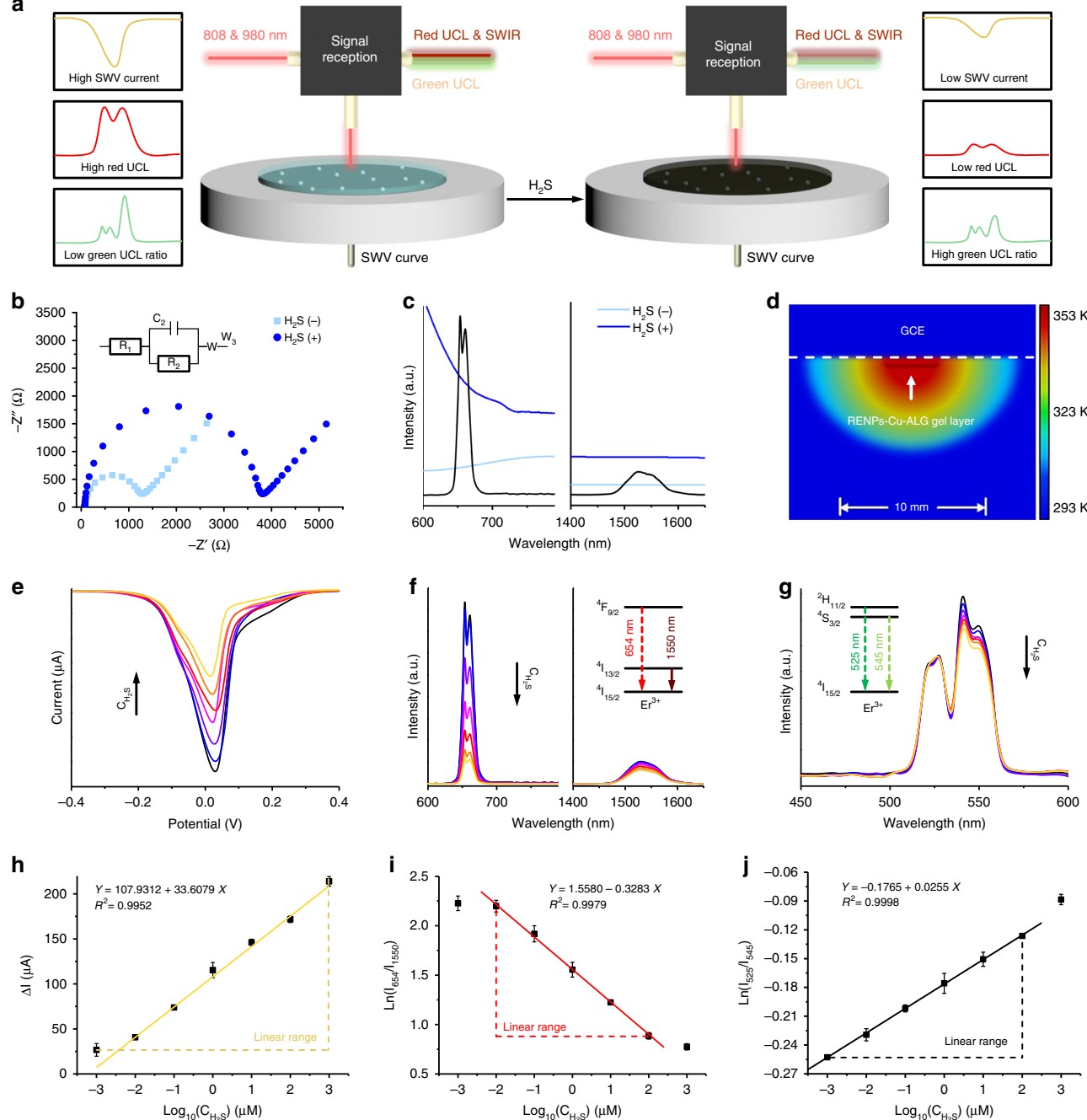

**Fig. 2** H$_2$S quantifying capacities of three-channel platform. **a** Schematic illustrations of the H$_2$S quantifying mechanism of the rationally constructed three-channel platform. Electrochemical impedance spectra of three-channel platform (**b**) and ultraviolet–visible–near-infrared spectra of RENPs–Cu–ALG gel (**c**) in the absence and presence of H$_2$S. **d** Finite element simulation of the heat conduction of electrode surface induced by the photothermal process. GCE was the abbreviation of glassy carbon electrode. SWV curves (**e**), luminescence spectra (**f**), and thermal-sensitive UCL curve (**g**) of three-channel platform in response to various C$_{H2S}$. Linear relationship between Δl (**h**), Ln(I$_{654}$/I$_{1550}$) (**i**), and Ln(I$_{525}$/I$_{545}$) (**j**) of three-channel platform and various C$_{H2S}$. Data are represented as mean ± SD (n = 3).

Furthermore, under laser irradiation at 808 nm, the temperature of the electrode surface increased according to the photothermal conversion effect of formed CuS (Supplementary Fig. 7). As the temperature on electrode could not be detected by traditional digital thermometer, it was reasonable to represent temperature by UCL ratio and construct relationship between $C_{H2S}$ and thermo-sensitive UCL of RENPs that was excited by 980-nm laser irradiation (Supplementary Fig. 8). However, due to the relatively low photothermal conversion efficiency and molar extinction coefficient of CuS at 980 nm (Supplementary Fig. 9), single 980-nm laser irradiation could not excite photothermal conversion capacity of CuS efficiently (Supplementary Fig. 10). It resulted in small temperature change in response to unit $H_2S$ concentration, which indicated that the joint use of 808-nm and 980-nm laser irradiation was necessary for sensitive photothermal quantification (Supplementary Fig. 11). For luminescent thermometry, the homogeneous microscopic temperature distribution around nanothermometer was critical to ensure the accuracy. A single-particle model established with the finite element method was used to simulate the possible temperature distribution around the surface of the electrode in solution under 808-nm laser irradiation (Fig. 2d). Whereas the macroscopic temperature of the surrounding solution declined significantly with increasing distance from the electrode surface, the microscopic temperature was distributed homogeneously in the RENP–Cu–ALG-gel layer, which suggested that the RENPs were heated evenly and therefore allowed the temperature to be determined precisely. Therefore, the electrochemical, luminescence, and photothermal conversion capacities of the RENP–Cu–ALG-gel-modified electrode were simultaneously altered in response to $H_2S$, allowing the simultaneous quantification of $H_2S$ by the three-signal channel described above.

Then, its quantification performance was tested in biological $C_{H2S}$ range, which was from nanomolar to micromolar level according to our previous findings[6]. The changes in current ($\Delta I$), the natural logarithm value of luminescence ratio ($Ln[I_{654}/I_{1550}]$), and $Ln[I_{525}/I_{545}]$ correlated linearly with $C_{H2S}$. Among all three methods of quantification, the electrochemical method had the broadest linear range (1 nM–1 mM) and the lowest limit of detection ($Y = 107.9312 + 33.6079 \times$, $R = 0.9952$; limit of detection: 1 nM; Fig. 2e, h). The luminescence method (10 nM–100 µM; $Y = 1.5580 – 0.3283 \times$, $R = 0.9979$; limit of detection: 8 nM; Fig. 2f, i) and the photothermal method (1 nM–100 µM; $Y = -0.1765 + 0.0255 \times$, $R = 0.9998$; limit of detection: 1 nM; Fig. 2g, j) also displayed outstanding capacities for quantifying $H_2S$ in similar concentration ranges. The modified RENP–Cu–ALG gel responded rapidly to the addition of $H_2S$, which allowed the rapid $H_2S$ quantification (< 2 min, Supplementary Fig. 12). We also noted that the three-channel platform had high selectivity for $H_2S$ among all other biosulfur compounds tested (typically biothiols and ions; Supplementary Fig. 13). Furthermore, due to the mechanism of $H_2S$ sensing, the biophosphorus compounds (typical adenosine phosphates and ions; Supplementary Fig. 14) and other cellular signal molecules (NO and CO; Supplementary Fig. 15) also possessed negative influence on all three-signal change. In addition, due to the use of orthogonal emissions, the temperature change would not have observable influence on luminescence quantification of $H_2S$, while the elevated absorbance would also not affect the results of photothermal quantification (Supplementary Fig. 16). It was noteworthy that our method can react with and quantify the total amount of all dissolved $H_2S$, ionic $HS^-$, and ionic $S^{2-}$, and also in high agreement with the result that obtained from standard protocol (Supplementary Fig. 17). Therefore, with an excellent linear range in the biological $C_{H2S}$ range, a rapid $H_2S$ response, an advanced $H_2S$ selectivity and a high accuracy, the three-channel platform

was applicable to the simultaneous electrochemical, luminescent, and photothermal quantification of plasma $C_{H2S}$.

**Highly precise serodiagnosis by multi-signal quantification**. In patients with AP, $H_2S$ synthetase is abnormally activated, resulting in elevated plasma $C_{H2S}$[55]. To study the efficiency of the three-channel platform in AP serodiagnosis, an AP mouse model with excessive plasma $C_{H2S}$ was established by caerulein-induced inflammation (Fig. 3a, b). After the establishment of the model, localized inflammation and edema were observed in the haematoxylin- and eosin-stained pancreas and lung sections (Fig. 3c). $H_2S$ also activated myeloperoxidase[56], and the myeloperoxidase activity was higher in the lungs of the AP mice than in those of the normal ones (Fig. 3d, e). The three-channel platform was then used for the simultaneous quantification of plasma $C_{H2S}$. Importantly, all three signal clearly indicated that the plasma $C_{H2S}$ was significantly higher in the AP mice (electrochemical method: $103.19 \pm 28.27$ µM; luminescent method: $89.72 \pm 25.33$ µM; photothermal method: $67.23 \pm 21.35$ µM) than in the normal ones (electrochemical method: $51.31 \pm 7.01$ µM; luminescent method: $27.88 \pm 9.08$ µM; photothermal method: $39.33 \pm 22.57$ µM; Fig. 3f–h). Therefore, the three-channel platform had great potential utility in the serodiagnosis of AP by identifying significant elevations in plasma $C_{H2S}$.

Traditionally, serodiagnosis based on the quantification of a bio-indicator is simply performed with a single method, which means that the precision of the serodiagnosis can be affected by various circumstances. Therefore, current molecule serodiagnosis suffers seriously from false-positive and negative results (Fig. 4a). To demonstrate the improvement in the precision of AP serodiagnosis afforded by the three-channel platform, a receiver-operating characteristic analysis was performed for each method, and the area under the characteristic curve (AUC) was used to evaluate their precision. Each of the single methods had normal serodiagnostic precision, with AUCs in the range of 0.795–0.940 (95% confidence interval), demonstrating that the false-positive/negative rate for the identification of AP exceeded 6%, which indicated an elevated risk of misdiagnosis (Fig. 4b–d). Similarly, the joint use of any two of these methods also incurred a relatively high false-positive/-negative rate (AUC < 0.941, false-positive/-negative rate > 5.9%; Fig. 4e–g). However, when the three-signal method was used, the calculation was complicated by the multiple potential thresholds. When the threshold was too low (threshold = 1/3) or too high (threshold = 1), the serodiagnosis precision was reduced (0.750–0.879, false-positive/negative rate > 12.1%), contrary to expectation, which was caused by the dramatic increase in the numbers of false-positive or false negative cases, respectively (Fig. 4h, j). When the threshold was optimized, the AUC was estimated from the characteristic curve to be 0.990 (95% confidence interval), indicating a 99% chance of correctly identifying cases of AP with the three-signal method, which was significantly higher than the AUCs for the single- or two-signal methods (Fig. 4i). Therefore, the serodiagnostic precision in the detection of AP was significantly improved by the three-signal method based on the three-channel platform with an optimized threshold.

## Discussion

To improve the precision of AP serodiagnosis by constructing a three-channel platform with accessional system for the quantification of serum $H_2S$, we modified a glassy carbon electrode with RENPs and Cu–ALG gel. The RENPs were ideal nanophosphors with multiple orthogonal emission (525 nm: $^2H_{11/2} \rightarrow {^4}I_{15/2}$, 545 nm: $^2S_{3/2} \rightarrow {^4}I_{15/2}$, 654 nm: $^4F_{9/2} \rightarrow {^4}I_{15/2}$ and 1550 nm: $^4I_{13/2} \rightarrow {^4}I_{15/2}$), relatively high thermal sensitivity (0.0108 K$^{-1}$) and low

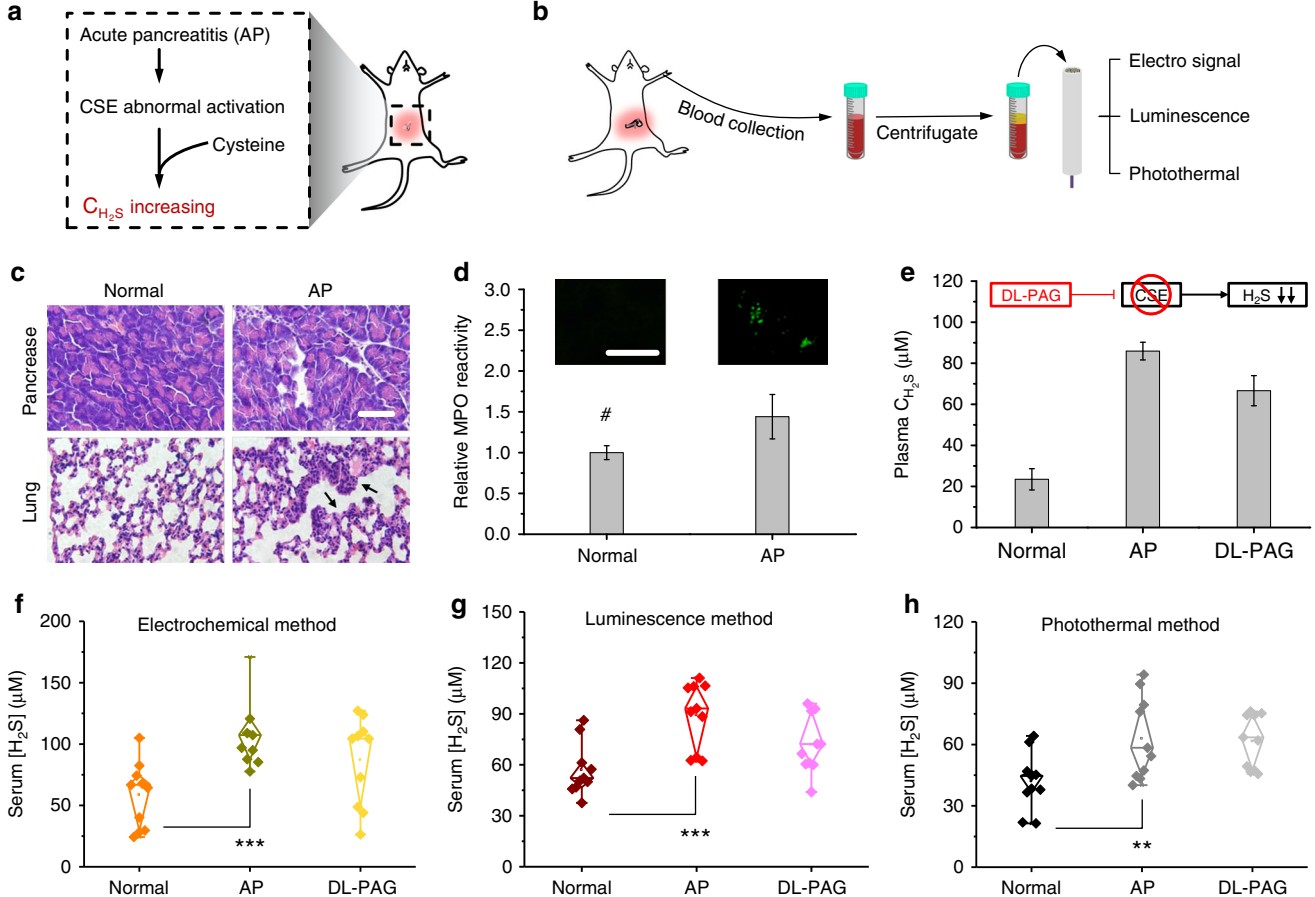

**Fig. 3** Evaluating plasma $H_2S$ level by three-channel platform. Schematic illustrations of in abnormal production of $H_2S$ in AP model (**a**) and plasma $H_2S$ quantifying experiments (**b**). **c** Hematoxylin- and eosin-stained sections of pancreases and lungs harvested from normal mice and AP mice. **d** Relative myeloperoxidase (MPO) activity in lungs and myeloperoxidase immunofluorescence stained lung sections (insert) of normal mice and AP mice. **e** Plasma $C_{H2S}$ of different groups sensed by standard colorimetric protocol. CSE cystathionine γ-lyase. Plasma $C_{H2S}$ of different groups simultaneously sensed by electrochemical (**f**), luminescence (**g**), and photothermal (**h**) method using three-channel platform. Data are represented as mean ± SD ($n = 3$). Statistical significance was determined from one-way $t$ tests. **$p < 0.01$ and ***$p < 0.001$. Scale bar, 50 μm.

temperature uncertainty (0.2559 K). The $Cu^{2+}$ in the gel rapidly (< 2 min) and specifically reacted with $H_2S$ to form insoluble semiconducting CuS in situ, which increased both the resistance of the gel and its visible/near-infrared absorbance. This increased resistance reduced the electrochemical signal in response to $H_2S$, allowing the electrochemical quantification. The elevated visible/near-infrared absorbance of the gel not only resulted in the selective quenching of red UCL of RENPs at 654 nm and the decrease of red UCL-to-SWIR ratio ($I_{654}/I_{1550}$) but also promoted the photothermal temperature increase under near-infrared laser irradiation, which was detectable by the ratio of thermally sensitive UCL ($I_{525}/I_{545}$) on the interface of the electrode in situ. Therefore, based on the rational combined use of three methods, a three-signal method for $H_2S$ quantification was established with outstanding performance from 1 nM to 1 mM. The three-channel platform could identify the difference in the plasma $C_{H2S}$ of AP and normal mice, which allowed effective identification of AP cases from normal ones by serodiagnosis. It should be noted that either single- or two-signal methods based on this platform were suffered from relatively low serodiagnosis precision (79.5–94.1%, lower than 95%), while the three-signal method with unreasonable threshold (1/3 or 1) was also unfavorable for precise serodiagnosis (75.0% and 87.9%, lower than 95%). However, by using the three-signal method and an optimized threshold for the three-channel platform, the precision of AP serodiagnosis was

significantly improved by 4.9%, which indicated that there was a 99% chance to identify AP cases correctly. Consequently, by focusing on the abnormal and much more specific biomolecular characteristics of serum or other biofluids, a highly precise serodiagnosis can be achieved in various diseases with a rationally designed multichannel platform. This should be of great interest to the medical industry. Our study not only provides a method with which to construct a multichannel platform for the simultaneous multi-signal quantification of a single bio-indicator, but also demonstrates a way to improve medical serodiagnostic precision.

## Methods

**Materials**. Rare-earth oxide $Lu_2O_3$ (99.999%), $Yb_2O_3$ (99.99%), and $Er_2O_3$ (99.99%) were purchased from STREM Chemicals, Inc. USA. HCl, NaOH, ethanol, and cyclohexane were purchased from Beijing Chemical Reagent Company. 1-Octadecene and $NH_4F$ were purchased from Alfa Aesar Chemical Co. Ltd. Alginic acid (sodium salt from brown algae, low viscosity), tricarbonylchloro(glycinato) ruthenium (II), sodium nitroferricyanide (III) dihydrate, and other chemicals were purchased from Sigma Aldrich. Rare-earth chlorides ($LnCl_3$, Ln: Lu, Yb, and Er) were prepared by dissolving the corresponding metal oxide in HCl solution at elevated temperature and then evaporating the water completely under reduced pressure. All other chemical reagents were of analytical grade and were used directly without further purification. Deionized water was used throughout.

**Synthesis of $NaYbF_4$:Er@$NaLuF_4$ nanoparticles (RENPs)**. In a typical experiment, a mixture of 1 mM $LnCl_3$ (Ln: 85% Yb and 15% Er), 15 mL oleic acid, and 15

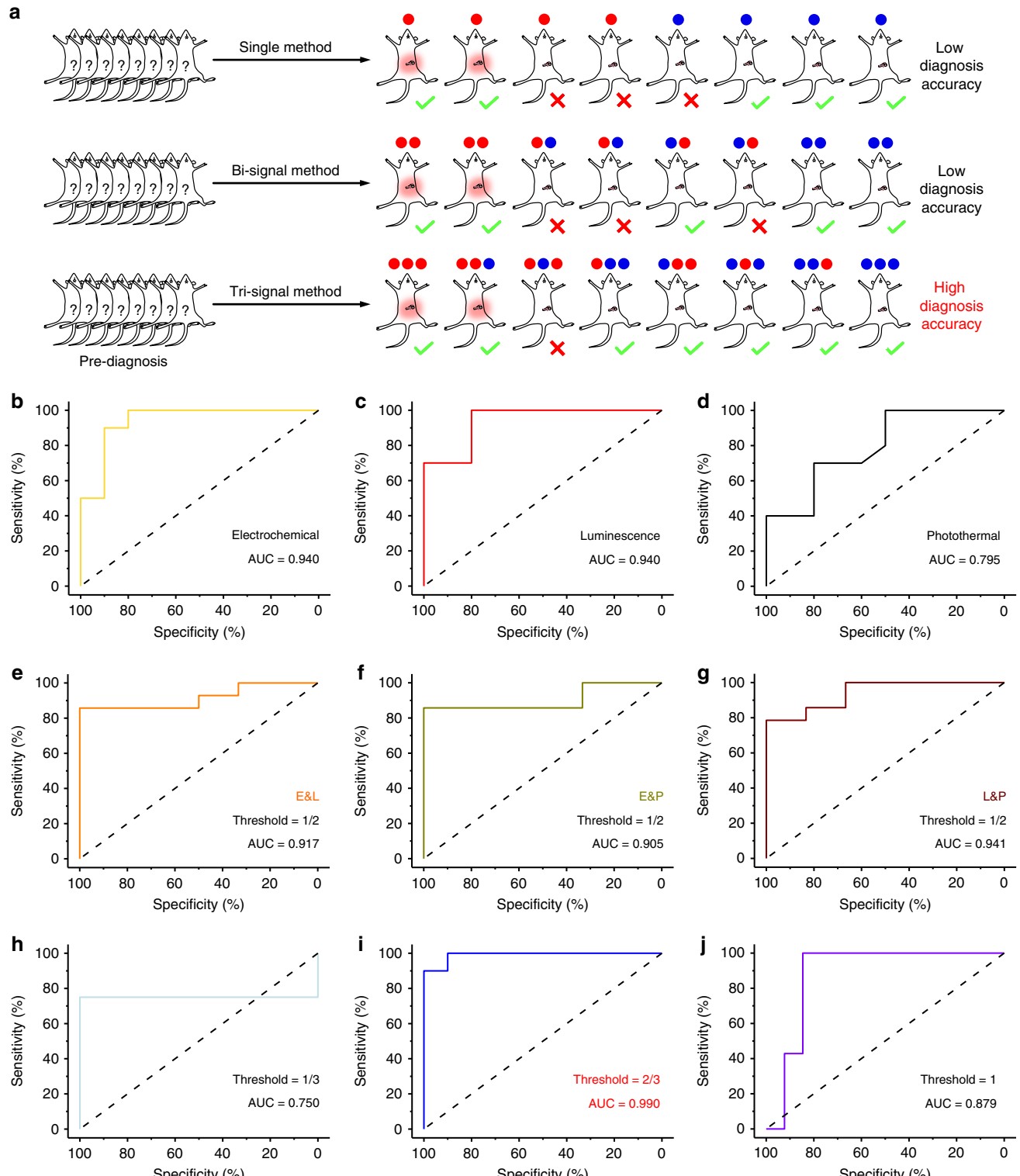

**Fig. 4** Improving precision serodiagnosis by three-channel platform. **a** Schematic illustrations of AP serodiagnosis by single-, two-signal and three-signal method based on three-channel platform. Receiver-operating characteristic curve presenting the probability for the assay to correctly distinguish between normal and AP cases based on the plasma $C_{H2S}$ in the blood samples determined by various methods including single-signal method by electrochemical (**b**), luminescence (**c**), and photothermal signal (**d**); two-signal method with joint use of electrochemical and luminescence (e&l, **e**), of electrochemical and photothermal (e&p, **f**), and of luminescence and photothermal (l&p, **g**); three-signal method with threshold of 1/3 (**h**), with threshold of 2/3 (**i**), and with threshold of 1 (**j**).

mL octadecene were added into a 100 mL three-necked flask. Under the vacuum, the mixture was heated to 160 °C to form a clear solution, and then cooled to room temperature. After the solution cooling down, 0.025 mmol NaOH and 0.04 mmol NH$_4$F were added into the flask directly and stirred for 30 min. The solution was slowly heated with gently stirred, degassed at 100 °C, and then heated to 300 °C and maintained for 1 h under the argon atmosphere. After the solution was cooled naturally, the NaYbF$_4$:Er nanoparticles were separated via centrifugation (9391 × $g$, 5 min) and washed with the mixture of ethanol and cyclohexane (1:1 v/v) for three times. The core-shell RENPs were obtained with the same solvothermal method using NaYbF$_4$:Er nanoparticles as core and LuCl$_3$ as only rare-earth source of inner shell. The RENPs were hermetically stored in cyclohexane under 4 °C.

**Constructing three-channel platform for H$_2$S quantification.** Glassy carbon electrodes were polished with Al$_2$O$_3$ powder, then ultrasonicated and washed by deionized water for three times. The construction was processed by modulating RENPs amount. In a typical experiment, 10 μL of ALG solution (0.2%) containing 1 mg mL$^{-1}$ RENPs was dropped on the surface of electrode and dried under 37 °C. Then 20 μL CuCl$_2$ solution (5 mM) was dropped on the electrode under 37 °C for 45 min. After washing extra CuCl$_2$ by deionized water, the electrode-based three-channel platform was successfully constructed and stored at dry conditions under room temperature.

**Characterization.** The sizes and morphologies of the nanoprobes were determined at 200 kV by a JEOL JEM-2010 transmission electron microscopy. The high-resolution lattice image and energy dispersive X-ray surface scan were performed using a Tecnai G$^2$F30 high-resolution transmission electron microscopy. Samples were dispersed in cyclohexane or deionized water, and dropped on the surface of a copper grid. The size distribution was counted and calculated from transmission electron microscopic images (α = 0.90, 500 particles were measured). The morphologies and energy dispersive X-ray surface scan of three-channel platform were determined by a Hitachi SU8010 scanning electron microscopy. Ultraviolet–visible-near-infrared absorbance spectra were obtained on a Shimadzu UV-3600 UV–vis–NIR spectrophotometer. Solid ultraviolet–visible–near-infrared diffuse reflectance spectra were obtained using a Hitachi U-4100 UV/Vis spectrophotometer. Powder X-ray diffraction pattern was measured with a Brucker D8 advance X-ray diffractometer from 10° to 70° (Cu Kα radiation, λ = 1.54 Å). Föurier transform infrared spectra were measured using a Shimadzu Föurier Transform Infrared Spectrophotometer IRPRESTIGE-21 from samples in KBr pellets. X-ray photoelectron spectra were performed on Thermo escalab 250Xi. The powder sample for determination was previously dried in nitrogen atmosphere at 120 °C. Electrochemical measurements were carried out by a Bio-Logic VMP3 multichannel potentiostat at ambient temperature. Square-wave voltammetry (SWV) curves were obtained by electrochemical station (Chenhua Instruments Co. CHI600). A three-electrode system was made up of a glassy carbon electrode (3 mm in diameter) as the working electrode, an Ag/AgCl electrode (saturated KCl) as a reference electrode, and a Pt wire as counter electrode. The UCL spectra were taken on a Maya LIFS-980 fluorescence spectrometer. The SWIR spectra were taken on an Ocean FLAME-NIR-INTSMA25 fluorescence spectrometer. The fluorescence spectrometer was united with electrochemical station to construct three-channel platform as Fig. 1a represented. The three-channel platform equipped with an external 0–8 W 980 nm and 808 nm adjustable lasers as the excitation source.

**Detection of H$_2$S in vitro.** Due to the neutral nature of serum, the H$_2$S would not only dissolve but ionize to be S$^{2-}$ and HS$^-$. To simulate the existence of H$_2$S in physiological conditions, H$_2$S standard solution was prepared by adjusting the pH of freshly formulated Na$_2$S solution to neutral (pH = 7.0) with diluted hydrochloric acid solution in medical saline. All H$_2$S standard solutions were hermetically stored at 4 °C and used within 60 min after formulation. The H$_2$S standard solutions were pre-quantified by using the standard methylene blue colorimetric protocol to ensure the accuracy of H$_2$S concentration for further study (Supplementary Fig. 18).

To set up the standard curve for H$_2$S detection, 10 μL of H$_2$S standard solutions with various final concentrations were incubated on the three-channel platform for 1 h, respectively. The electrochemical measurement was conducted from −0.4 to 0.4 V in saline (0.1 M, pH = 7.4) with pulse amplitude of 25 mV and an increase potential of 4 mV s$^{-1}$. Meanwhile, the three-channel platform received merged irradiation of 808 nm (1.5 W cm$^{-2}$) and 980 nm (0.75 W cm$^{-2}$) lasers. The luminescence signals were collected simultaneously.

Other biosulfurs compounds, biophosphorus compounds, and other cellular signal molecules were also studied by the same method to demonstrate the selectivity of three-channel platform to H$_2$S, including reduced glutathione (GSH), cysteine (Cys), SO$_4^{2-}$, SO$_3^{2-}$, S$_2$O$_3^{2-}$, adenosine triphosphate (ATP), adenosine diphosphate (ADP), adenosine monophosphate (AMP), pyrophosphoric acid (PPi), nitric monoxide (NO), and carbon monoxide (CO). The concentration of studied biosulfurs compounds, biophosphorus compounds, and cellular signal molecules was 100 mM. H$_2$S standard solution (1 mM) was used for comparison.

**Mechanism study of H$_2$S detection.** To obtain the products in reaction, 1 mM H$_2$S standard solution was incubated on Cu–ALG gel for 1 h. The products of reaction were collected via filtration. All the above products were washed with deionized water for several times, lyophilized under nitrogen protection, and stored in individual sealed container before further determination, respectively. The characterization of studied products were performed within 24 h post preparation.

In neutral condition, H$_2$S exists as the forms of dissolved H$_2$S, ionic HS$^-$ and ionic S$^{2-}$. According to the determination results, the following chemical equation was proposed to describe the reaction between Cu–ALG gel and H$_2$S in neutral condition:

$$Cu - ALG + H_2S = CuS + H_2 - ALG \tag{1}$$

$$Cu - ALG + HS^- = CuS + H - ALG^- \tag{2}$$

$$Cu - ALG + S^{2-} = CuS + ALG^{2-} \tag{3}$$

**Dynamic study of H$_2$S detection.** The amount of CuS formed in solid Cu–ALG gel was monitored in the dark room by ultraviolet–visible–near-infrared diffuse reflectance spectra. In a typical experiment, 10 μL of H$_2$S standard solution (1 mM) was dropped on gel and monitored for 2 min continuously. To optimize incubation time for electrochemical method, the H$_2$S standard solution was incubated on three-channel platform with incubation time increasing from 20 to 40 min.

**Ethical statement.** The establishment of AP mice was performed by Beijing Vital River Laboratory Animal Technology Co., Ltd. in accordance with the Guidelines for Care and Use of Laboratory Animals and approved by the Animal Ethics Committee of the Vital River Institutional Animal Care and Use Committee.

**AP modal establishment.** To establish the AP mice modal, C57BL/6 mice were intraperitoneally injected with caerulein solution (50 μg kg$^{-1}$) six times, and were continuously observed within the following 72 h. The successful establishment of AP mice modal was confirmed by corresponding H&E-stained pancreas sections from randomly selected mice. The myeloperoxidase immunostained lung sections were also performed to demonstrate the AP-associated lung injury. The serum H$_2$S concentration was determined by typical colorimetric assay.

**Serodiagnosis of AP by three-channel platform.** AP mice were randomly divided into 2 groups ($n = 10$, m = 20 ± 5 g) for further study. Ten AP mice in negative control group were intraperitoneally injected with H$_2$S-synthetse inhibitor DL-propargylglycine (DL-PAG, 450 μmol kg$^{-1}$) per 3 h for four times (DL-PAG group). The normal mice were used as blank control. The serum of mice was collected from the femoral vein and separated by centrifugation. Another ten AP mice and ten normal mice were used for receiver-operating characteristic analysis.

**Statistical analysis.** The three-signal method was used to analyze ten serum samples collected from normal mice and ten serum samples collected from AP mice. The statistical test was two-sided. Receiver-operating characteristic curve analysis was utilized to determine the AP serodiagnosis potential that the ability to correctly differentiate normal cases and AP cases by three-signal method. The receiver-operating characteristic curve analysis captures the trade-off between sensitivity and specificity while changing a discrimination threshold, but it can be summarized as a single measurement. The sensitivity (true positive rate) was plotted against the specificity (true negative rate) in the characteristic curve as a function of a variety of thresholds of class prediction probabilities. The overall accuracy depends on the output signal distributions for the two classes, in this case, normal mice and AP mice. Values range between 0.5 and 1.0, where a value of 0.5 indicates that the two distributions are identical, and a value of 1.0 indicates that there is no overlap in the distributions of output signals for the two classes. The AUC is used as a lone measure of evaluating the efficiency of the model ranked subjects according to the probability assigned to the positive class. The AUC was calculated by the trapezoidal method of integration with the corresponding 95% confidence intervals. The single- and two-signal method were also studied for comparison.

**Reporting summary.** Further information on research design is available in the Nature Research Reporting Summary linked to this article.

## Data availability
The data that support the findings of this study are available from the authors on reasonable request.

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

## Acknowledgements

Y.L. personally thank Dr. Xingjun Zhu from Stanford University for intensive discussion in preparation of response and Dr. Luoyuan Li from Tsinghua University for determination of thermal-responsive luminescent spectrum. Z.W. personally thank Yun Zheng from University of Macau for suggestion in electrochemical determination. The authors thank the funding of National Natural Science Foundation of China (21673143), Natural Science Foundation of Beijing Municipality (2172016, KZ201810028045), Beijing

Municipal Education Commission Outstanding Young Individual Project (CIT&TCD201904082), Beijing Talent Foundation Outstanding Young Individual Project (2015000026833ZK02), Capacity Building for Sci-Tech Innovation-Fundamental Scientific Research Funds (19530050179, 025185305000/195), Yanjing Young Scholar Program of Capital Normal University, Project of High-level Teachers in Beijing Municipal Universities in the Period of 13th Five-year Plan (IDHT20180517) and Beijing Advanced Innovation Centre for Imaging Theory and Technology.

## Author contributions

Y.L. and Z.W. contributed equal to this work. Z.M. and J.Z. supervised the project. All authors conceived the project, analyzed the results, and wrote the original paper. Z.W. and Y.L. synthesized the RENPs and constructed the three-channel platform. Z.W. performed electrochemical quantifying experiments. Y.L. performed luminescence and photothermal quantifying experiments.

## Competing interests

The authors declare no competing interests.
