## [Peer Review File · Nature Communications]

Reviewers' comments:

Reviewer #1 (Remarks to the Author):

The manuscript deals with a new strategy for H₂S detection by the three-channel platform for the simultaneous three-signal quantification of H₂S by modifying a glassy carbon electrode with a luminescent rare-earth-based nanothermometer and Cu-alginate (Cu-ALG) gel. The performance of this three-channel platform in quantifying H₂S was studied in vitro and in vivo using an established AP mouse model. The study is interesting and informative in the field of H₂S probes. However, it may not be acceptable for publication in nature communications. There are several matters that has to be addressed by the authors before considering to resubmit to another journal.

1. H₂S detection by using this tri-channel platform is an irreversible process, so it does not have an advantage, how this system can be improved or modified to make it reversible? Reversible detection systems are more advantageous as compared to irreversible system.
2. Detection by using copper complexes have been known for ATP ADP AMP and PPI sensor. Therefore, authors need to do more experiment to find out the interference of ATP, ADP AMP and PPI.
3. For the selectivity studies of the tri-channel platform, the author also should test the response with other cellular signal molecules NO and CO.
4. How to confirm the H₂S presence in solution as discussed on page 15 line 321-324, After the addition of Na₂S in water. The Na₂S with water change to SH⁻ and NaOH. And it is not stable, but it is kept in a tri-channel platform for 1 hr. Did the authors confirm the H₂S in test solution by using any other method?
5. The reference section has to be checked and modified according to the journal format. For example see reference 8, 10, 14, 16, and 19.
6. Title of the manuscript is not clear and it does not reflect the merit of the work, it has to be modified.
7. Title of the article in the manuscript file and SI file are different, it has to be corrected both should be same.

Reviewer #2 (Remarks to the Author):

Manuscript ID: NCOMMS-19-19045

Title: Simultaneous Multi-signal Quantification of a Single Bio-indicator Improves the Precision of Serodiagnosis

Authors: Yuxin Liu, Zheng Wei, Jing Zhou, Zhanfang Ma

General Comments: Liu & Wey et al report the development of a three-channel H₂S quantification platform constructed on a glassy carbon electrode using core-shell NaYbF₄:Er@NaLuF₄ nanoparticles. The system is designed and constructed to simultaneously measure the luminescence emission spectrum and the electrochemical signal of the probe with which the electrode modified by introducing an electrochemical workstation, visible and infrared luminescence emission spectroscopy. The Ln³⁺-doped nanoparticles synthesis and characterization are presented namely the size distribution. The photophysical emission of the prepared platform are studied in the presence of H₂S. The use of dual irradiation wavelengths allow simultaneous excitation of the Ln³⁺-doped nanoparticles (by the 980 nm laser) and the photothermal effect due to the metallic electrode (by the 808 nm laser, particularly) and thus the authors combine the distinct readouts to improve the H₂S quantification by combinatorial analysis of the three outputs. The novelty of this work is the incorporation of three outputs that resulted in an improved precision of acute pancreatitis serodiagnosis. In general, the paper is not reasonably well written however with too many abbreviations, repetition of the same ideas and

unbalanced references to the relevant literature. The manuscript length is appropriate for this journal, although the presented figures are not adequate and deserve some effort looking forward to improving their clarity. One weak point of the work is the quantification of the temperature increase resulting (essentially) from the 808 nm radiation absorption readout and that can be quantified using the visible emission of the Er³⁺ ion, however the strong points are the novelty in the combination of 3 signals for H₂S quantification and acute pancreatitis serodiagnosis in animal models. In summary, the paper may be adequate for Nature Communications Journal if the authors can address the major concerns detailed next. In the present form this reviewer's recommendation is to accept it for publication after major revisions.

Particular Comments:

1. The text is very repetitive, and the same idea is repeated in every section thus the text flowing is very difficult. It is mandatory that the authors revise the text to eliminate the over repetition of the same ideas.
2. The overuse of abbreviations makes the text reading very difficult. In the perspective of a broad audience journal like Nature Communications this is not adequate because it results in an additional difficulty, especially for the non-experienced readers. Please consider reducing it to the minimum possible.
3. The references need to include some works of researchers in distinct latitudes. The list of references is biased towards Asiatic authors when the field presents important contributions from European and American groups. Please consider including them.
4. The major concern of this reviewer is the quantification of H₂S towards the emission lines of Er³⁺ in the visible spectral range. These transitions are known to being used for luminescence thermometry and thus it is absolutely necessary to include the temperature raise resulting separately by the 980 nm, by the 808 and 980 nm irradiation (laser power densities are missing) and the cross effect between the temperature and the H₂S quantification.
5. The luminescent thermometer performance parameters (thermal sensitivity and temperature uncertainty) need to be calculated and compared with the literature values.
6. The conclusions are quite qualitative so some numerical discussion is needed to give some strength to the main points focused in the work.

Reviewer #1

General comments: *The manuscript deals with a new strategy for H₂S detection by the three-channel platform for the simultaneous three-signal quantification of H₂S by modifying a glassy carbon electrode with a luminescent rare-earth-based nanothermometer and Cu-alginate (Cu-ALG) gel. The performance of this three-channel platform in quantifying H₂S was studied in vitro and in vivo using an established AP mouse model. The study is interesting and informative in the field of H₂S probes. However, it may not be acceptable for publication in nature communications. There are several matters that has to be addressed by the authors before considering to resubmit to another journal.*

Comment 1: *H₂S detection by using this tri-channel platform is an irreversible process, so it does not have an advantage, how this system can be improved or modified to make it reversible? Reversible detection systems are more advantageous as compared to irreversible system.*

Response: Here, we discussed and studied the potential of reversibility of our as-developed tri-channel platform. According to the mechanism of H₂S response, the formation of CuS principally contributes to the change of all three signals. Therefore, to achieve reversion of the tri-channel platform, the elimination of CuS was priority, so we explored the methods and protocols to decompose CuS. It was found that the

solution of oxidants can oxidize this insoluble sulfide into soluble sulphate (Please see Response Figure 1A), which could decompose and eliminate CuS by washing-out (Please see Response Figure 1B). As a result, it would be applicable to eliminate CuS from electrode, which preliminarily allowed the reversion of the tri-channel platform.

Response Figure 1. Decomposition of produced CuS on H₂S-treated Cu-ALG gel by solution of oxidants. The absorbance at 808 nm A) and X-ray diffraction patterns B) of H₂S-treated Cu-ALG gel treated with acidic H₂O₂ solution (Cu-ALG + H₂S + H₂O₂ + H⁺) and with HNO₃ solution (Cu-ALG + H₂S + HNO₃). The H₂S-treated Cu-ALG gel and un-treated Cu-ALG gel were used as positive or negative control. Data are represented as mean ± SD.

Though the elimination of CuS from electrode is likely to be achieved based on the pilot study shown above, it should be noted that, however, there are still some questions that have to be concerned for the reversion of the electrode due to the intrinsic limitation of range of biomedical applications, which was not only originated from our work but from most of previously researches and reported diagnostic systems (Gorbet, M. *et al.* Hemostasis and thrombosis: basic principles and clinical practice. Colman, R. Eds.; JB Lippincott: Philadelphia, PA, 2006; 751-760. Mrksich,

M. *et al.* Peptide chips for the quantitative evaluation of protein kinase activity. *Nat. Biotechnol.* **2002**, *20*, 270-274). First of all, unlike those sensors and systems for only detection *in vitro*, the actual biological samples for diagnostic platform were biological fluids, which has complicated chemical composition. There were many macromolecules, such as proteins and nucleic acids, in the biological fluids, which would adsorb on the platform during quantification process. To achieve the actual reversion of diagnostic platform, these macromolecules must be removed from platform, which required sophisticated procedures and expensive reagents (proteinases or nucleases). Considering the low-cost and facile preparation procedure of our tri-channel platform, it was much more practical and easier to administrate new ones rather than reverse them. In addition, the platform for disease diagnosis should be clean and disposable to ensure the safety and reduce cross-infection. The re-use of reversed platform will result in the elevation of risk of infections, which was unacceptable for actual diagnosis applications. As a result of the above-mentioned questions, most of advanced platform for diagnosis, efficient and clinically applicable though they were, were irreversible (Hassibi, A. *et al.* Multiplexed identification, quantification and genotyping of infectious agents using a semiconductor biochip. *Nat. Biotechnol.* **2018**, *36*, 738-745; Johnson, K. *et al.* Semisynthetic sensor proteins enable metabolic assay at the point of care. *Science* **2018**, *361*, 1122-1126; Heath, J. *et al.* Integrated barcode chips for rapid, multiplexed analysis of proteins in microliter quantities of blood. *Nat. Biotechnol.* **2008**, *36*, 1373-1378). Therefore, though we believed that reversible diagnostic system may have advantage, it was still an

insoluble issue in current design and construction of diagnostic system due to the imbalance between economic benefits and medical reliability.

Meanwhile, we would like to iterate that the primary goal of this work is to propose a novel strategy to improve serodiagnosis precision by reducing false positive/negative rate *via* multi-signal quantification. To achieve this, a tri-channel platform was designed for H₂S quantification by rationally optimizing probes and choosing signal types. By this multi-channel platform, the precision of AP diagnosis was significantly improved from 75.0-94.1% (single or bi-signal method) to 99% (tri-signal method). Instead of optimizing the sensor for H₂S sensing alone, we attempted to propose a novel and convenient strategy to improve diagnostic precision. We appreciated the reviewer for this suggestion and would further optimize our device in our future works.

Comment 2: *Detection by using copper complexes have been known for ATP ADP AMP and PPI sensor. Therefore, authors need to do more experiment to find out the interference of ATP, ADP AMP and PPI.*

Response: We appreciate this valuable advice. We tested the response of our tri-channel platform with biophosphorus compounds, including ATP, ADP, AMP and PPI (Please see Response Figure 2 and Supplementary Figure 14). Though some works demonstrated the application of copper complexes in luminescent sensors of biophosphorus compounds, the as-developed tri-channel platform possesses insignificant response to all these studied biophosphorus compounds. It can be

explained by the difference of mechanism between gel and previous reported sensors. Traditionally, the copper complexes were used as energy acceptors in constructing luminescent sensor for sensing of biophosphorus compounds by quenching luminescence through Fröster resonance energy transfer (FRET) route, which occurred within only nanoscale (< 10 nm) (Ouyang, J. et al. Plasmon-enhanced fluorescence-based core-shell gold nanorods as a near-IR fluorescent turn-on sensor for the highly sensitive detection of pyrophosphate in aqueous solution. *Adv. Funct. Mater.* **2015**, *25*, 7017; Li, Z. et al. Nitrogen-doped carbon dots mediated fluorescent on-off assay for rapid and highly sensitive pyrophosphate and alkaline phosphatase detection. *Sci. Rep.* **2017**, *7*, 5849). In the presence of biophosphorus compounds, the Cu^{2+} in acceptors complexed with negative phosphorus-contained groups, resulting in the reduction of FRET and hence luminescence recovery. However, in our cases, the luminescence of RENPs are not quenched by copper complex through FRET route due to the microscale distance between them, which indicated that the luminescence of RENPs won't recover in the presence of biophosphorus. Meanwhile, in the presence of H_2S , CuS formed on gel *in situ* with strong near-infrared absorbing, which selectively quenched the red luminescence of RENPs through re-absorption route, which allowed the ratiometric luminescence quantification of H_2S . Furthermore, unlike H_2S , all studied biophosphorus (such as ATP, ADP, AMP and PPi) cannot react with gel and form products with both high resistance and photothermal conversion capacities. Therefore, typical biophosphorus won't affect the quantification of H_2S .

Response Figure 2 and Supplementary Figure 14. Selectivity of tri-channel platform to H₂S among biophosphors. ΔI (A), absorbance (B), $\text{Ln}(I_{654}/I_{1550})$ (C), and $\text{Ln}(I_{525}/I_{545})$ (D) of tri-channel platform in response to H₂S (10 μM) and other biophosphorus in 100 times higher concentration (1 mM). The studied typical biophosphorus included adenosine triphosphate (ATP), adenosine diphosphate (ADP), adenosine monophosphate (AMP) and pyrophosphoric acid (PPi). Data are represented as mean \pm SD.

Related discussions were added in the Results and Methods section in the revised manuscript (Please see Line 2-6, Page 9 and Line 17-19, Page 16). Data were added in the revised supplementary files (Please see Supplementary Figure 14).

Comment 3: *For the selectivity studies of the tri-channel platform, the author also should test the response with other cellular signal molecules NO and CO.*

Response: We appreciate this valuable advice. To further illustrate the selectivity of the tri-channel platform to H₂S, we further studied the response with other cellular

signaling molecules, including NO and CO (Please see Response Figure 3 and Supplementary Figure 15). It was found that all three signals barely changed in response to either NO or CO, which could be explained by the mechanism of gel for H₂S response: In the presence of H₂S, CuS formed on gel *in situ* with capabilities of all high resistance, strong near-infrared absorbing and outstanding photothermal conversion, which thence led to the simultaneous change of all electrochemical, luminescence and photothermal signals. Unlike H₂S, both NO and CO cannot react with gel and form products with the above three capacities. Therefore, cellular signal molecules like NO and CO won't possess influence on the quantification of H₂S.

Response Figure 3 and Supplementary Figure 15. Selectivity of tri-channel platform to H₂S among other cellular signal molecules. ΔI A), absorbance B), $\ln(I_{654}/I_{1550})$ C), and $\ln(I_{525}/I_{545})$ D) of tri-channel platform in response to H₂S (10 μM) and other cellular signal molecules in 100 times higher concentration (1 mM). The studied typical cellular signal molecules included NO and CO. Data are represented as mean \pm SD.

Related discussions were added in the Results and Methods section in the revised manuscript (Please see Line 2-6, Page 9 and Line 17-19, Page 16). Data were added in the revised supplementary files (Please see Supplementary Figure 15).

Comment 4: *How to confirm the H₂S presence in solution as discussed on page 15 line 321-324, After the addition of Na₂S in water. The Na₂S with water change to SH- and NaOH. And it is not stable, but it is kept in a tri-channel platform for 1 hr. Did the authors confirm the H₂S in test solution by using any other method?*

Response: We appreciate this valuable advice. To study the stability of test solution and confirm the reliability of the quantification experiments, we quantify the H₂S concentration of as-prepared Na₂S solution by using the standard methylene blue colorimetric protocol within 100 min (Kevil, C. et al. Hydrogen sulfide chemical biology: Pathophysiological roles and detection. *Nitric Oxide-Biol. Chem.* **2013**, *35*, 5), a longer time than our studied period of 60 min (1 h). The test solution was freshly prepared and hermetically stored at 4 °C. It was found that the total H₂S concentration kept stable within the studied period (Please see Response Figure 4 and Supplementary Figure 17), which indicated that the H₂S concentration of test solution will be reliable if prepared freshly and used timely.

Response Figure 4 and Supplementary Figure 17. C_{H₂S} uncertainty of standard H₂S solution for *in vitro* study. A) Linear relationship between various C_{H₂S} and absorbance at 665 nm that determined by typical colorimetric protocol. B) C_{H₂S} change as a function of storage time within 100 min post-preparation. Theoretical C_{H₂S} was also marked as reference. Data are represented as mean ± SD. It was found that the C_{H₂S} of standard H₂S solution only slightly fluctuated around 50 μM within 100 min, which suggested that the concentration of H₂S standard solution was accurate if prepared freshly and used timely. It also ensured the reliability of C_{H₂S} within incubation time of 1 h. Data are represented as mean ± SD.

However, it was previously demonstrated that the H₂S mainly existed as S²⁻ and HS⁻ in aqueous biological system due to the neutral nature of serum (Sen, U. et al. Increased endogenous H₂S generation by CBS, CSE, and 3MST gene therapy improves ex vivo renovascular relaxation in hyperhomocysteinemia. *Am. J. Physiol.-Cell Ph.* **2012**, 303, 41; Yoo, J. et al. Immobilization of the gas signaling molecule H₂S by radioisotopes: detection, quantification, and in vivo imaging. *Angew. Chem. Int. Ed.* **2016**, 128, 9511). Therefore, to simulate the existence of H₂S in psychological conditions, the H₂S standard solution (pH = 7.4) contains all dissolved H₂S, ionic HS⁻ and ionic S²⁻, but no NaOH. It was noteworthy that our tri-signal method can react with and quantify the total amount of all dissolved H₂S, ionic HS⁻ and ionic S²⁻, and also in high agreement with the result that obtained from standard

methylene blue colorimetric protocol (Please see Response Figure 5 and Supplementary Figure 16).

Response Figure 5 and Supplementary Figure 16. Comparison between tri-signal method and standard colorimetric method for H₂S detection. C_{H₂S} determined by electrochemical method A), luminescence method B) and photothermal method C). Corresponding C_{H₂S} was previously determined by standard colorimetric method and used for comparison. D) Data statistic of the above determined C_{H₂S} by various methods. The above results illustrated that the C_{H₂S} determined by all three methods were highly in accordance with that by standard method, which suggested that all three methods were applicable for accurate quantification of H₂S. Data are represented as mean ± SD. Statistical significance was determined from one-way t tests. N.S. means not significant.

In addition, it should be noted that the incubation time of 60 min (1 h) was a combination of the time for reaction and the time for signal stabilization (Please see

Response Figure 6). H_2S (also including HS^- and S^{2-}) can react with ALG-Cu gel rapidly (< 2 min), which can be confirmed by the dynamic ultraviolet-visible-near infrared spectra (Please see Response Figure 7A and Supplementary Figure 12A). Therefore, the H_2S in solution can be consumed in a short period, which means that the change of H_2S concentration within long-term (> 100 min) may not possess significant influence on quantification result. Though the reaction period is short, the time for stabilizing signal varies due to different mechanism of signal change (Please see Response Figure 7B-D and Supplementary Figure 12B-D): 1) For electrochemical signal, the H_2S contribute to both elimination of electrical signal substances and growth of semiconductor CuS *in situ*, which simultaneously decrease electrical signal and increase resistance, synthetically resulting in change of electrochemical signal; 2) For luminescence signal, the grown CuS selectively absorbed the luminescence of RENPs in red spectral region but no SWIR spectral region, which allowed the change of luminescence ratio in these two wavelength range; 3) For photothermal signal, the grown CuS convert laser irradiation into heat by non-radiative relaxation, heating the RENPs and resulting in change of thermal-sensitive luminescence. Therefore, the incubation time of 60 min (1 h) was a comprehensive consideration to ensure the stabilization of all three signals instead of finishing reaction between gel and H_2S .

In summary, according to results obtained by using standard methylene blue colorimetric protocol, H_2S concentration of test solution will be reliable within the studied period of 60 min (1 h). Furthermore, though H_2S ionized in biological aqueous system and existed in ionized state of HS^- and S^{2-} , it would not influence the

quantification results as our tri-signal method was able to effectively quantify all kinds of H₂S and in high accordance with standard colorimetric protocol.

Response Figure 6. Schematic illustration and comparison of reaction time, signal stabilization time and H₂S concentration reliable time. It was obvious that H₂S concentration reliable time (100 min) is longer than the combination of both reaction time and signal stabilization time (as known as incubation time of 60 min), which indicated that the H₂S concentration was reliable within the incubation period.

Response Figure 7 and Supplementary Figure 12. Response rate of the tri-channel platform to H₂S. Absorbance at 808 nm A), current B), $\text{Ln}(I_{654}/I_{1550})$ C) and $\text{Ln}(I_{325}/I_{545})$ D) of

RENPs-Cu-ALG gel in response to H₂S as a function of time. Data are represented as mean ± SD.

Related discussions were added in the Results and Methods section in the revised manuscript (Please see Line 6-9, Page 9; Line 22, Page 15; Line 1-7, Page 16 and Line 7-12, Page 17). Data were added in the revised supplementary files (Please see Supplementary Figure 12, Supplementary Figure 16 and Supplementary Figure 17).

Comment 5: *The reference section has to be checked and modified according to the journal format. For example see reference 8, 10, 14, 16, and 19.*

Response: We appreciate this kind remind. We have checked the reference section according to the journal format. For example,

Reference 8: “Kolanowski, J. L., Liu, F. & New, E. J. Fluorescent probes for the simultaneous detection of multiple analytes in biology. *Chem. Soc. Rev.* **47**, 195-208, doi:10.1039/c7cs00528h (2018)” was corrected to be “Kolanowski, J. L. *et al.* Fluorescent probes for the simultaneous detection of multiple analytes in biology. *Chem. Soc. Rev.* **47**, 195-208, doi:10.1039/c7cs00528h (2018)”.

Reference 10: “Goyaa, M. E., Romanowski, A., Caldart, C. S., Benard, C. Y. & Golombek, D. A. Circadian rhythms identified in caenorhabditis elegans by in vivo long-term monitoring of a bioluminescent reporter. *P. Natl. Acad. Sci. USA* **113**, 7837-7845, doi:10.1073/pnas.1605769113 (2016)” was corrected to be “Goyaa, M. E. *et al.* Circadian rhythms identified in caenorhabditis elegans by in vivo long-term monitoring of a bioluminescent reporter. *P. Natl. Acad. Sci. USA* **113**, 7837-7845,

doi:10.1073/pnas.1605769113 (2016)”;

Reference 14 “Quesada-Gonzalez, D. & Merkoci, A. Nanomaterial-based devices for point-of-care diagnostic applications. *Chem. Soc. Rev.* **47**, 4697-4709, doi:10.1039/c7cs00837f (2018)” was corrected to be “Quesada-Gonzalez, D. *et al.* Nanomaterial-based devices for point-of-care diagnostic applications. *Chem. Soc. Rev.* **47**, 4697-4709, doi:10.1039/c7cs00837f (2018)”.

Reference 16 and 19 were replaced by other pioneer works according to the request of Reviewer 2 and were also checked.

In addition, the whole reference list was checked and updated according to the format of Nature Communications.

Comment 6: *Title of the manuscript is not clear and it does not reflect the merit of the work, it has to be modified.*

Response: We appreciate this valuable advice. In this work, we were proposing a strategy to reduce the false positive/negative rate of serodiagnosis and hence achieve higher precision by a multi-signal quantification method. For this purpose, a multi-channel platform was rationally constructed. To verify the practicability, acute pancreatitis was chosen as disease model and H₂S was used as indicator for serodiagnosis. Therefore, based on the merit of this work, we changed the title from “Simultaneous Multi-signal Quantification of a Single Bio-indicator Improves the Precision of Serodiagnosis” to “Simultaneous Multi-signal Quantification for Highly Precise Serodiagnosis Utilizing Rationally Constructed Platform”. We hope it can

clearly demonstrate our novelty and main findings.

Comment 7: *Title of the article in the manuscript file and SI file are different, it has to be corrected both should be same.*

Response: We appreciate this kind remind. The title of SI file has been corrected in accordance with that of manuscript file. We apologize for this unintentional mistake.

Reviewer #2:

General comments: *Liu & Wey et al report the development of a three-channel H₂S quantification platform constructed on a glassy carbon electrode using core-shell NaYbF₄:Er@NaLuF₄ nanoparticles. The system is designed and constructed to simultaneously measure the luminescence emission spectrum and the electrochemical signal of the probe with which the electrode modified by introducing an electrochemical workstation, visible and infrared luminescence emission spectroscopy. The Ln³⁺ doped nanoparticles synthesis and characterization are presented namely the size distribution. The photophysical emission of the prepared platform are studied in the presence of H₂S. The use of dual irradiation wavelengths allow simultaneous excitation of the Ln³⁺-doped nanoparticles (by the 980 nm laser) and the photothermal effect due to the metallic electrode (by the 808 nm laser, particularly) and thus the authors combine the distinct readouts to improve the H₂S quantification by combinatorial analysis of the three outputs. The novelty of this work is the incorporation of three outputs that resulted in an improved precision of acute pancreatitis serodiagnosis. In general, the paper is not reasonably well written however with too many abbreviations, repetition of the same ideas and unbalanced references to the relevant literature. The manuscript length is appropriate for this journal, although the presented figures are not adequate and deserve some effort looking forward to improving their clarity. One weak point of the work is the quantification of the temperature increase resulting (essentially) from the 808 nm radiation absorption readout ant that can be quantified using the visible emission of*

the Er³⁺ ion, however the strong points are the novelty in the combination of 3 signals for H₂S quantification and acute pancreatitis serodiagnosis in animal models. In summary, the paper may be adequate for Nature Communications Journal if the authors can address the major concerns detailed next. In the present form this reviewer's recommendation is to accept it for publication after major revisions.

Comment 1: *The text is very repetitive, and the same idea is repeated in every section thus the text flowing is very difficult. It is mandatory that the authors revise the text to eliminate the over repetition of the same ideas.*

Response: We appreciate this valuable advice. We tried our best to remove the repetitive sentences and ideas. For example, the following paragraphs were removed.

“To demonstrate the potential utility of the RENP-Cu-ALG-modified GCE as a platform for the simultaneous three-signal determination of H₂S, the determination mechanism was studied” (Please see Line 16-18, Page 6 in the original manuscript).

“Therefore, as C_{H₂S} increased, the current signal and red UCL intensity decreased, whereas the photothermal temperature increased, corresponding to the CuS formed on the three-channel platform (Fig. 2D–F)” (Please see Line 5-8, Page 8 in the original manuscript).

“To ascertain the difference in plasma C_{H₂S} between the AP and normal mice, the plasma C_{H₂S} was first determined with a standard colorimetric kit (Fig. 3E). The plasma C_{H₂S} in the AP mice was clearly higher than that in the normal mice, confirming that elevated plasma C_{H₂S} characterized the mice with AP. Therefore, the

AP mouse model, with excessive plasma C_{H_2S} , was successfully established for further study” (Please see Line 12-17, Page 9 in the original manuscript)

In addition, the following sentences were simplified.

“Therefore, the rational construction of an efficient multichannel platform is vital if we are to improve the precision of medical serodiagnostics. To develop a novel multichannel platform, a rationally selected quantification method with good applicability and associativity is required” (Please see Line 3-6, Page 3 in the original manuscript) were simplified to be “Therefore, the rational construction of an efficient multichannel platform by combining applicable quantification method is vital” (Please see Line 4-5, Page 3 in the revised manuscript).

“Therefore, by modifying the electrode with a rare-earth-based nanothermometer and an H_2S -responsive electro-active material that can also act as a luminescence acceptor and a photothermal conversion agent, a multichannel platform can be established to simultaneously determine electrochemical, optical, and thermal signals for H_2S quantification” (Please see Line 11-15, Page 4 in the original manuscript) was simplified to be “As a result of the above theoretical basis, a multichannel platform could be established for simultaneously multi-signal quantification” (Please see Line 11-12, Page 4 in the revised manuscript).

“To demonstrate the potential utility of the RENP-Cu-ALG-modified GCE as a platform for the simultaneous three-signal determination of H_2S , the determination mechanism was studied” (Please see Line 16-18, Page 6 in the original manuscript) was simplified to be “After confirming successful construction, the determination

mechanism of platform was then studied” (Please see Line 18-19, Page 6 in the revised manuscript).

“A detailed evaluation was conducted to assess the quantification performance of the as-developed three-channel platform. According to our previous findings, the biological $C_{\text{H}_2\text{S}}$ range is from the nanomolar to the micromolar level” (Please see Line 3-5, Page 8 in the original manuscript) was simplified to be “Then, its quantification performance was tested in biological $C_{\text{H}_2\text{S}}$ range, which is from nanomolar to micromolar level according to our previous findings” (Please see Line 11-12, Page 8 in the revised manuscript).

Therefore, 3 repetitive paragraphs have been removed and 4 sentences have been simplified. We hope the text was smooth to read after cutting down redundancy.

Comment 2: *The overuse of abbreviations makes the text reading very difficult. In the perspective of a broad audience journal like Nature Communications this is not adequate because it results in an additional difficulty, especially for the non-experienced readers. Please consider reducing it to the minimum possible.*

Response: We appreciate this valuable advice. We deleted the unnecessary abbreviations and reduced them to the minimum possible. For example, most of the abbreviations of instruments and materials were deleted, including high resolution transmission electron microscopy, transmission electron microscopy, powder X-ray diffraction patterns, energy dispersive X-ray analysis, scanning electronic microscopy, electrochemical impedance spectroscopy, Föurier transform infrared spectrum, X-ray

photoelectron spectroscopy, glassy carbon electrode, 1-octadecene, oleic acid and deionized water. Moreover, the abbreviations of those proper nouns that presented in manuscript less than 5 times were also removed, including ultraviolet-visible-near infrared, near-infrared, finite element method, limit of detection, haematoxylin and eosin, receiver operating characteristic curve and Vital River Institutional Animal Care and Use Committee. Therefore, more than 60 abbreviations have been removed and only those important retained in the manuscript. We hope it could be adequate for both professional and non-experienced readers of Nature Communications.

Comment 3: *The references need to include some works of researchers in distinct latitudes. The list of references is biased towards Asiatic authors when the field presents important contributions from European and American groups. Please consider including them.*

Response: We appreciate this kind remind. It was an unintentional mistake rather than bias that we cited more works from Asiatic authors than that from European and American groups. We apologize for our carelessness. To prevent this misunderstanding, we consulted a number of papers about this subject and added advanced pioneering works from European and American groups that initiatively studying luminescent nanoparticles or functionalized electrodes to quantify biomolecules and temperature. Therefore, 17 references have been added or replaced to keep the balance of reference. We hope it could prevent the misunderstanding.

(1) Prof. Daniel Jaque from Universidad Autónoma de Madrid (Spain) made

outstanding contribution to the field of various luminescent nanoparticles in luminescent thermometry and comprehensively highlighted their potential in various bioapplications.

- Reference 26: Martín Rodríguez, E. *et al.* Persistent luminescence nanothermometers. *Appl. Phy. Lett.* **2017**, *111*, 081901;
- Reference 49: del Rosal, B. *et al.* Infrared-emitting QDs for thermal therapy with real-time subcutaneous temperature feedback. *Adv. Funct. Mater.* **2016**, *26*, 6060-6068;
- Reference 50: Labrador-Paez, L. *et al.* Reliability of rare-earth-doped infrared luminescent nanothermometers. *Nanoscale* **2018**, *10*, 22319-22328;
- Reference 51: Carrasco, E. *et al.* Intratumoral thermal reading during photo-thermal therapy by multifunctional fluorescent nanoparticles. *Adv. Funct. Mater.* **2015**, *25*, 615-626;

(2) Prof. Pavlos G. Lagoudakis from University of Southampton (UK) analyzed and modulated the dye-sensitized upconversion luminescence in depth.

- Reference 39: Alyatkin, S. *et al.* In-depth analysis of excitation dynamics in dye-sensitized upconversion core and core/active shell nanoparticles. *J. Phys. Chem. C*, **2018**, *122*, 18177-18184;

(3) Prof. Markus Haase from Universität Osnabrück (Germany) and Prof. Artur Bednarkiewicz from Institute of Temperature and Structure Research, Polish Academy of Science (Poland) proposed strategy for improving upconversion luminescence quantum yield.

- Reference 41: Homann, C. *et al.* NaYF₄:Yb,Er/NaYF₄ core/shell nanocrystals with high upconversion luminescence quantum yield. *Angew. Chem. Int. Ed.* **2018**, *57*, 8765-8769;
 - Reference 39: Pilch, A. *et al.* Shaping luminescent properties of Yb³⁺ and Ho³⁺ co-doped upconverting core-shell β-NaYF₄ nanoparticles by dopant distribution and spacing. *Small*, **2017**, *13*, 1701635;
- (4) Prof. Niko Hildebrandt from Université Paris-Sud (France) studied the FRET between upconversion luminescent donor and acceptors.
- Reference 46: Bhuckory, S. *et al.* Core or shell? Er³⁺ FRET donors in upconversion nanoparticles. *Eur. J. Inorg. Chem.* **2017**, *17*, 5186-5195;
 - Reference 44: Cardoso Dos Santos, M. *et al.* Recent developments in lanthanide-to-quantum dot FRET using time-gated fluorescence detection and photon upconversion. *TrAC-Trend. Anal. Chem.* **2016**, *84*, 60-71;
- (5) Prof. Parak Wolfgang from Philipps-Universität Marburg (Germany) and Prof. Loïc J. Charbonnière from Université de Strasbourg (France) explored the application of rare-earth-based nanomaterials in biosensing.
- Reference 45: Escudero, A. *et al.* Rare earth based nanostructured materials: Synthesis, functionalization, properties and bioimaging and biosensing applications. *Nanophotonics* **2017**, *6*, 881;
 - Reference 17: Sy, M. *et al.* Lanthanide-based luminescence biolabelling. *Chem. Commun.* **2016**, *52*, 5080-5095;
- (6) Prof. Ekaterina I. Galanzha from University of Arkansas for Medical Sciences

(USA), Prof. Mostafa El-Sayed from University of California at San Francisco (USA) and Prof. Paresh Chandra Ray from Jackson State University (USA) used photothermal signal as a tool to quantify biomolecules.

- Reference 13: Nedosekin, D. *et al.* Photoacoustic and photothermal detection of circulating tumor cells, bacteria and nanoparticles in cerebrospinal fluid in vivo and ex vivo. *J. Biophotonics*, **2013**, *6*, 523-533;
- Reference 18: Jain, P. *et al.* Noble metals on the nanoscale: Optical and photothermal properties and some applications in imaging, sensing, biology, and medicine. *Acc. Chem. Res.* **2008**, *41*, 1578-1586;
- Reference 19: Viraka Nellore, B. *et al.* Aptamer-conjugated theranostic hybrid graphene oxide with highly selective biosensing and combined therapy capability. *Faraday Discuss.* **2014**, *175*, 257-271;

(7) Prof. Miguel de la Guardia from University of Valencia (Spain) reviewed recent advance of electrochemical biosensing.

- Reference 15: Karimzadeh, A. *et al.* Electrochemical biosensing using N-GQDs: Recent advances in analytical approach. *TrAC-Trend. Anal. Chem.* **2018**, *105*, 484-491;

(8) Prof. Georgios A. Sotiriou from ETH Zurich (Switzerland) contributed to the luminescent sensing of biomolecules.

- Reference 16: Pratsinis, A. *et al.* Enzyme-mimetic antioxidant luminescent nanoparticles for highly sensitive hydrogen peroxide biosensing. *ACS Nano* **2017**, *11*, 12210-12218;

(9) Prof. Michael R. Hamblin from Harvard University (USA) constructed upconversion nanovehicle for drug delivery.

- Reference 40: Karimi, M. *et al.* Smart nanostructures for cargo delivery: Uncaging and activating by light. *J. Am. Chem. Soc.* **2017**, *139*, 4584-4610.

Comment 4: *The major concern of this reviewer is the quantification of H₂S towards the emission lines of Er³⁺ in the visible spectral range. These transitions are known to being used for luminescence thermometry and thus it is absolutely necessary to include the temperature raise resulting separately by the 980 nm, by the 808 and 980 nm irradiation (laser power densities are missing) and the cross effect between the temperature and the H₂S quantification.*

Response: We appreciate this valuable advice. In our work, the mechanism of photothermal quantification of H₂S can be described as the following processes: 1) Cu²⁺ in gel reacted with H₂S, forming typical photothermal conversion agent CuS; 2) Under 808 nm laser irradiation (1.5 W cm⁻²), CuS generated heat and increased surrounding temperature of RENPs; 3) Under 980 nm laser irradiation (0.75 W cm⁻²), RENPs emitted green upconversion luminescence (Please see Response Figure 8 and Figure 1H), which changed in response to temperature change (Please see Response Figure 9A and Supplementary Figure 8A). Therefore, H₂S concentration can be correlated with the ratio of thermal-sensitive emission of Er³⁺, which allowed the luminescence-assisted temperature sensing and therefore photothermal quantification of H₂S on electrode *in situ* (Please see Response Figure 9B-E and Supplementary

Figure 8B-E).

Response Figure 8 and Figure 1. Characterization of tri-channel platform. A) Schematic presentation of as-designed tri-channel platform based on RENPs-Cu-ALG gel-modified electrode. transmission electron microscopic image B) and energy dispersive X-ray element surface scan C) of RENPs. D) A plot of $\ln(I_{525}/I_{545})$ versus $1/T$ to calibrate the thermometric scale for RENPs. Scanning electron microscopic image E) and energy dispersive X-ray

element surface scan F) of tri-channel platform. SWV curve G), green UCL H), red UCL I), and SWIR luminescence J) spectrum of tri-channel platform.

Response Figure 9 and Supplementary Figure 8. Constructing relationship between C_{H_2S} and $\text{Ln}(I_{525}/I_{545})$. A) Signal transition process from C_{H_2S} to $\text{Ln}(I_{525}/I_{545})$. Linear relationship between C_{CuS} and C_{H_2S} B), temperature and $\text{Log}_{10}(C_{CuS})$ C), temperature and $\text{Log}_{10}(C_{H_2S})$ D) and $\text{Ln}(I_{525}/I_{545})$ and reciprocal of temperature ($1/T$) E). Through facile algebraic operation, linear relationship can be constructed between C_{H_2S} and $\text{Ln}(I_{525}/I_{545})$ of RENPs-based nanothermometer, which allow the photothermal quantification of H_2S by determining photothermal-induced temperature change of electrode *in situ*. Data are represented as mean \pm SD.

To further understand the influence of laser irradiation on photothermal quantification and luminescence thermometry, temperature raise resulted by single

808 nm laser irradiation, by single 980 nm laser irradiation and by dual laser irradiations were studied and compared. It was found by heating curves that, under single 808 nm laser irradiation for 15 s, the H₂S-treated gel had higher temperature raise (6.38 °C) than those without H₂S treatment (1.04 °C), which indicated that the photothermal conversion capacities of gel were activated by H₂S and hence reasonable to construct relationship between temperature and H₂S concentration (Please see Response Figure 10 and Supplementary Figure 7). Furthermore, it was also noticed that the 980 nm laser cannot raise temperature efficiently (1.49 °C) under the studied power density (Please see Response Figure 11 and Supplementary Figure 10), which might be a combined result of following factors including decreasing molar extinction coefficient ($\epsilon = 5.64 \times 10^9 \text{ mol}^{-1} \text{ cm}^{-1}$ at 980 nm and $6.71 \times 10^9 \text{ mol}^{-1} \text{ cm}^{-1}$ at 808 nm) and photothermal conversion efficiency of H₂S-treated gel at various spectral wavelengths ($\eta = 7.2\%$ at 980 nm and 18.1% at 808 nm), as well as the different laser power density used in this study (0.75 W cm^{-2} for 980 nm laser irradiation and 1.5 W cm^{-2} for 808 nm laser irradiation, Please see Response Figure 12 and Supplementary Figure 9). Due to these differences, the temperature raise by co-use of 808 nm and 980 nm laser was similar to that by single 808 nm laser irradiation, demonstrating that the 980 nm laser won't significantly influence the temperature rising (Please see Response Figure 13 and Supplementary Figure 11). More importantly, the co-use of 808 nm and 980 nm laser irradiation can simultaneously activate efficient photothermal conversion process and strong thermal-sensitive luminescence, generating heat and heating NaYbF₄:Er@NaLuF₄,

which was able to cause obvious change of luminescence ratio for photothermal quantification of H₂S concentration.

Based on the above findings, it was rational to use 808 nm laser irradiation to activate photothermal conversion capacities of H₂S-activated gel and use 980 nm laser irradiation to activate thermal-sensitive emission of NaYbF₄:Er@NaLuF₄ for temperature monitoring, further allowing the indirectly photothermal quantification of H₂S concentration by thermal-sensitive luminescence.

Response Figure 10 and Supplementary Figure 7. H₂S-activated photothermal capacities of RENPs-ALG-Cu gel under single 808 nm laser irradiation (1.5 W cm⁻²). A) Temperature curves of gel in the presence and absence of H₂S. B) Linear time data vs -Ln(Theta) obtained from the cooling period. C) Photothermal images of gel in the presence (upper) and absence of H₂S (bottom) under single 808 nm laser irradiation within 30 s.

Response Figure 11 and Supplementary Figure 10. H₂S-activated photothermal capacities of RENPs-ALG-Cu gel under single 980 nm laser irradiation (0.75 W cm⁻²). A) Temperature curves of gel in the presence and absence of H₂S. B) Linear time data vs -Ln (Theta) obtained from the cooling period. C) Photothermal images of gel in the presence (upper) and absence of H₂S (bottom) under single 980 nm laser irradiation within 30 s.

Response Figure 12 and Supplementary Figure 9. Comparison of molar extinction coefficient (ϵ) and photothermal conversion efficiency (η) of RENPs-ALG-Cu gel at various wavelengths. Absorbance at 808 nm A) and at 980 nm B) of gel in the presence of H_2S . ϵ C) and η D) of H_2S -treated gel at various wavelengths. Data are represented as mean \pm SD.

Response Figure 13 and Supplementary Figure 11. H_2S -activated photothermal capacities of RENPs-ALG-Cu gel under dual 808 nm (1.5 W cm^{-2}) and 980 nm laser irradiation (0.75 W cm^{-2}). A) Photothermal images of gel in the presence (upper) and absence of H_2S (bottom) under dual 808 nm and 980 nm laser irradiation within 30 s. B) Temperature curves of gel in the presence and absence of H_2S . C) $\ln(I_{525}/I_{545})$ curves of gel in the presence of H_2S under various laser irradiation prescriptions. Data are represented as mean \pm SD.

Related discussions were added in the Results and Methods section in the revised manuscript (Please see Line 11-22, Page 7). Data were added in the revised manuscript and supplementary files (Please see Figure 1D, Supplementary Figure

7-11).

Comment 5: *The luminescent thermometer performance parameters (thermal sensitivity and temperature uncertainty) need to be calculated and compared with the literature values.*

Response: We appreciate this valuable advice. We determined the performance parameters of NaYbF₄:Er@NaLuF₄-based luminescent thermometer, referring to thermal sensitivity and temperature uncertainty, to evaluate their capability in temperature sensing. First of all, the ratio of green UCL of NaYbF₄:Er@NaLuF₄ at 525 nm and 545 nm, Ln(I₅₂₅/I₅₄₅), was determined under various temperature to obtain a calibration for temperature sensing (Please see Response Figure 8 and Figure 1D). It was found that Ln(I₅₂₅/I₅₄₅) possessed good linear relationship with 1/T (273 K-353 K; Y = 1.9672 – 0.6411 X, R = 0.9969), which confirmed the capability of accurate temperature sensing on electrode *in situ* by luminescence.

The thermal sensitivity and temperature uncertainty was then determined and calculated according to an advanced pioneering work (Jaque, D. et al. Infrared-emitting QDs for thermal therapy with real-time subcutaneous temperature feedback. *Adv. Funct. Mater.* **2016**, *26*, 6060). The thermal sensitivity of RENPs was calculated by a standard equation for ratiometric luminescent nanothermometers:

$$S = \frac{1}{R} \frac{dR}{dT} \quad \text{Supplementary Equal 1}$$

where S is thermal sensitivity, R is I₅₂₅/I₅₄₅ luminescence ratio and T is temperature in Kelvin unit. The thermal sensitivity of RENPs was calculated to be 0.0108 K⁻¹, which

was comparable to most rare-earth-based luminescent nanothermometers (Please see Response Figure 14A and Supplementary Figure 3A, Response Table 1 and Supplementary Table 1).

Response Figure 14 and Supplementary Figure 3. Thermo-sensitive capacities of RENPs.

Thermal sensitivity A) and temperature uncertainty of RENPs-based luminescent nanothermometer.

Response Table 1 and Supplementary Table 1. Comparison of thermal sensitivity of RENPs with other previously reported rare-earth-based luminescent nanothermometers.

Nanothermometers	Thermal sensitivity (K^{-1})	Reference
$\text{NaYbF}_4:\text{Er}@/\text{NaLuF}_4$	0.0108	This work
$\text{CaF}_2:\text{Yb,Er}$	0.0160	ACS Nano 2011 , 5, 8665
$\text{NaLuF}_4:\text{Yb,Er}@/\text{NaLuF}_4$	0.0100	Nat. Commun. 2016 , 7, 1047
$\text{Y}_2\text{O}_3:\text{Yb,Ho,Zn}$	0.0100	Dalton. Trans. 2013 , 42, 11005
$\text{LaF}_3:\text{Nd}$	0.0026	Adv. Funct. Mater. 2015 , 25, 615
$\text{CaF}_2:\text{Yb,Tm}$	0.0020	ACS Nano 2011 , 5, 8665

Response Figure 15 and Figure 2. H₂S quantifying capacities of tri-channel platform. A) Schematic illustrations of the H₂S quantifying mechanism of the rationally constructed tri-channel platform. Electrochemical Impedance Spectra of tri-channel platform B) and ultraviolet-visible-near infrared spectra of RENPs-Cu-ALG gel C) in the absence and presence of H₂S. D) Finite element simulation of the heat conduction of electrode surface induced by the photothermal process. SWV curves E), luminescence spectra F), and thermal-sensitive UCL curve F) of tri-channel platform in response to various C_{H_2S} . Linear relationship between ΔI G), $\text{Ln}(I_{654}/I_{1550})$ H), and $\text{Ln}(I_{525}/I_{545})$ I) of tri-channel platform and various C_{H_2S} . Data are represented as mean \pm SD.

The temperature uncertainty of RENPs was calculated from experimental data included in Response Figure 15 and Figure 2G by an equation in first order approximation:

$$\delta T = \frac{1}{S} \frac{\Delta I}{I} \quad \text{Supplementary Equal 2}$$

where δT is temperature uncertainty, S is thermal sensitivity, ΔI is background noise of luminescence spectrum and I is luminescence intensity. The temperature uncertainty of RENPs was calculated to be 0.2559 K (Please see Response Figure 14B and Supplementary Figure 3B), which was less than 1 K and hence applicable to quantify temperature according to previous reports (Jaque, D. et al. Ag/Ag₂S nanocrystals for high sensitivity near-infrared luminescence nanothermometry. *Adv. Funct. Mater.* **2017**, 27, 1604629).

Related discussions were added in the Results and Methods section in the revised manuscript (Please see Line 16-22, Page 5 and Line 1-5, Page 6). Data were added in the revised manuscript and supplementary files (Please see Supplementary Figure 3 and Supplementary Table 1).

Comment 6: *The conclusions are quite qualitative so some numerical discussion is needed to give some strength to the main points focused in the work.*

Response: We appreciate this valuable advice. We rewrote the conclusions and added numerical discussions to give strength to the main points of this work. The following sentences were added. We hope that these quantitative discussions can give strength to the main points focused in the work.

“The RENPs were rational nanophosphors with multiple orthogonal emission (525 nm: $^2H_{11/2} \rightarrow ^4I_{15/2}$, 545 nm: $^2S_{3/2} \rightarrow ^4I_{15/2}$, 654 nm: $^4F_{9/2} \rightarrow ^4I_{15/2}$ and 1550 nm: $^4I_{13/2} \rightarrow ^4I_{15/2}$), relatively high thermal sensitivity (0.0108 K^{-1}) and low temperature uncertainty (0.2559 K). The Cu^{2+} in the gel rapidly ($< 2 \text{ min}$) and specifically reacted with H_2S to form insoluble semiconducting CuS *in situ*, which increased not only the resistance of the gel but its visible/near-infrared absorbance.”

“It should be noted that either single- or two-signal methods based on this platform were suffer from relatively low serodiagnosis precision (79.5-94.1%, lower than 95%), while the three-signal method with unreasonable threshold (1/3 or 1) was also unfavourable for precise serodiagnosis (75.0% and 87.9%, lower than 95%). However, by using the three-signal method and an optimized threshold for the three-channel platform, the precision of AP serodiagnosis was significantly improved by 4.9%, which indicated that there was a 99% chance to identify AP cases correctly.”

Reviewers' comments:

Reviewer #1 (Remarks to the Author):

I am pleased that the authors considered the comments from the reviewers and revised the manuscript. The additional experiments and corrections made the manuscript in a better view. There are few mistakes that have to be corrected before recommending publication.

Figure 2J caption is missing

Figure 2E should be SWV curves

Figure 1B and 1C caption need to be corrected.

Reviewer #2 (Remarks to the Author):

Manuscript ID: NCOMMS-19-19045

Title: Simultaneous Multi-signal Quantification of a Single Bio-indicator Improves the Precision of Serodiagnosis

Authors: Yuxin Liu, Zheng Wei, Jing Zhou, Zhanfang Ma

General Comments: Liu & Wey et al presented the revised version of the manuscript. The authors present significant changes to the original manuscript by incorporating most of the changes and attending to the comments of both the referees. After revision, the paper English was significantly improved, and references were carefully revised for better balance between Asiatic and non-Asiatic authors. The figures were modified to improving their clarity. The quantification of the temperature increase resulting (essentially) from the 808 nm radiation absorption is now properly described. In summary, the paper was significantly improved with respect to the original version, and, in this reviewer opinion, it is adequate for Nature Communications Journal if the authors address the major concerns of both referees. In the present form this reviewer's recommendation is to accept it for publication after minor revisions.

Particular Comments:

1. A particular point on the comments on the references about the relative thermal sensitivity and on the temperature uncertainty. The concept of relative thermal sensitivity was originally proposed by Wade et al. in the context of optical fiber temperature sensors and proposed as a figure of merit for comparing thermometers, irrespectively of their nature for the first time in a review work of 2012 from the group of prof. L. Carlos.
2. The major concern of this reviewer is the quantification of H₂S using the emission lines of Er³⁺ in the visible spectral range (the same used for luminescence thermometry). The temperature raise resulting separately by the 980 nm and by the 808 nm irradiation (laser power densities are still missing). The temperature effect on the H₂S quantification is not discussed and needs to be adressed.
3. The luminescent thermometer performance parameters (thermal sensitivity and temperature uncertainty) were calculated but were not compared with the literature values.

Reviewer #1

General comments: *I am pleased that the authors considered the comments from the reviewers and revised the manuscript. The additional experiments and corrections made the manuscript in a better view. There are few mistakes that have to be corrected before recommending publication.*

Response: We appreciate the reviewer for the positive opinions on our revised manuscript.

Comment 1: *Figure 2J caption is missing. Figure 2E should be SWV curves. Figure 1B and 1C caption need to be corrected.*

Response: We appreciate these valuable advices. The caption of Figure 2J is added (Please see Line 7-10, Page 28). The sentences that referenced Figure 2E-J is corrected according to figures with one-to-one correspondence (Please see Line 20-22, Page 8 and Line 1-4, Page 9). The captions of Figure 1B, 1C and 1F are corrected in accordance with main text (Please see Line 4-7, Page 27).

Reviewer #2:

General comments: *Liu & Wey et al presented the revised version of the manuscript. The authors present significant changes to the original manuscript by incorporating most of the changes and attending to the comments of both the referees. After revision, the paper English was significantly improved, and references were carefully revised for better balance between Asiatic and non-Asiatic authors. The figures were modified to improving their clarity. The quantification of the temperature increase resulting (essentially) from the 808 nm radiation absorption is now properly described. In summary, the paper was significantly improved with respect to the original version, and, in this reviewer opinion, it is adequate for Nature Communications Journal if the authors address the major concerns of both referees. In the present form this reviewer's recommendation is to accept it for publication after minor revisions..*

Response: We appreciate the reviewer for the affirmation of our revised manuscript.

Comment 1: *A particular point on the comments on the references about the relative thermal sensitivity and on the temperature uncertainty. The concept of relative thermal sensitivity was originally proposed by Wade et al. in the context of optical fiber temperature sensors and proposed as a figure of merit for comparing thermometers, irrespectively of their nature for the first time in a review work of 2012 from the group of prof. L. Carlos.*

Response: We appreciate this valuable advice. According to your suggestion, we noticed that some of works that originally enlightened the thermal sensitivity and

temperature uncertainty domain was missed in the reference. Therefore, some more pioneer works including the above two are added into the reference list.

(49) Carlos, L. *et al.* Thermometry at the nanoscale. *Nanoscale* **2012**, *4*, 4799-4829;

(50) Wade, S. *et al.* Nd³⁺-doped optical fiber temperature sensor using the fluorescence intensity ratio technique. *Rev. Sci. Instrum.* **1999**, *70*, 4279-4282;

(51) Brites, C. *et al.* Ratiometric highly sensitive luminescent nanothermometers working in the room temperature range. Applications to heat propagation in nanofluids. *Nanoscale* **2013**, *5*, 7572-7580.

Comment 2: *The major concern of this reviewer is the quantification of H₂S using the emission lines of Er³⁺ in the visible spectral range (the same used for luminescence thermometry). The temperature raise resulting separately by the 980 nm and by the 808 nm irradiation (laser power densities are still missing). The temperature effect on the H₂S quantification is not discussed and needs to be addressed.*

Response: We appreciate this valuable advice. With two or more orthogonal luminescence emission, rare-earth-based upconversion nanoprobe are feasible for simultaneous detection of multiple bioindicators. In this work, we use green emission of RENPs (centred at 525 nm and 545 nm) for thermal sensing and photothermal quantification of H₂S, while utilizing red emission (centred at 654 nm) for luminescence quantification of H₂S by quenching mechanism of inner filter effect. Moreover, SWIR emission of RENPs (centred at 1550 nm) is applied as internal reference for luminescence quantification. Therefore, the emission for photothermal

and luminescence quantification of H₂S is orthogonal.

In order to evaluate the effect of temperature on luminescence quantification of H₂S, we thence studied the relationship between luminescent ratio ($\text{Ln}(I_{654}/I_{1550})$) and $C_{\text{H}_2\text{S}}$ in the absence of 808 nm laser irradiation (Please see Response Figure 1A). It is noticed that, in the absence of heating, a linear relationship is also found between $\text{Ln}(I_{654}/I_{1550})$ and $C_{\text{H}_2\text{S}}$ while the function ($Y = 1.5688 - 0.3295 X$) is similar to that in the presence of heating ($Y = 1.5580 - 0.3283 X$, Please see Figure 2I in revised manuscript). This phenomenon can be a result of relatively ultralow thermal sensitivity of red emission in steady-state luminescence spectrum.

Response Figure 1 and Supplementary Figure 16. Orthogonality between photothermal quantification and luminescence quantification method. A) Linear relationship between $\text{Ln}(I_{654}/I_{1550})$ of three-channel platform and various $C_{\text{H}_2\text{S}}$ in absence of 808 nm laser irradiation. B) Linear relationship between $\text{Ln}(I_{525}/I_{545})$ of three-channel platform and various temperature in presence of H₂S. Data are represented as mean \pm SD.

We further discussed the effect of elevated absorbance on temperature sensing capacities of RENPs by green emissions (Please see Response Figure 1B). It is found that the $\text{Ln}(I_{525}/I_{545})$ possessed linear relationship with $C_{\text{H}_2\text{S}}$ in the presence of H₂S while the function ($Y = 2.0624 - 0.6739 X$) is similar to that in the absence of H₂S (Y

= 1.9672 – 0.6411 X, Please see Figure 1D in revised manuscript). It can be explained by the mechanism of thermal-sensitive luminescence of Er³⁺: the increased temperature induced re-distributions in the populations of energy levels, further resulting in the change of luminescence ratio between 545 nm and 525 nm, which is ruled by Boltzmann distribution and is less relevant to absolute luminescence intensity. Therefore, though the elevated absorbance might result in some luminescence quenching, the luminescence ratio would not be affected significantly.

Based on the above findings, the photothermal and luminescence quantification method is orthogonal, which indicated that the above two methods, though both utilizing the emissions of Er³⁺ in visible spectral range, will not affect the quantification each other's result.

Laser power densities of 808 nm and 980 nm were added in the Methods section in the revised manuscript (Please see Line 21-22, Page 16). Related discussions were added in the Results section in the revised manuscript (Please see Line 12-15, Page 9). Data were added in the revised manuscript and supplementary files (Please see Supplementary Figure 16).

Comment 3: *The luminescent thermometer performance parameters (thermal sensitivity and temperature uncertainty) were calculated but were not compared with the literature values.*

Response: We appreciate this valuable advice. Estimating from the UCL signal changes in response to temperature changes, RENPs were good luminescent

nanothermometers with a relatively high sensitivity of 0.0108 K^{-1} and low temperature uncertainty of 0.2559 K around $273\text{-}353 \text{ K}$, which was comparable to most previously reported rare-earth-based nanothermometer and was enough to quantify temperature accurately (Please see Response Table 1). Though some of previous reported molecule thermometers showed more advanced temperature resolution and sensitivity (*Nature* **2013**, *500*, 54; *Angew. Chem. Int. Ed.* **2013**, *52*, 11154), they relied on single band emission without an internal reference, which may be affected by complex biological samples. Therefore, with adequate ratiometric luminescent thermometry performance, it is reasonable to use RENPs for sensitive and accurate temperature sensing.

Related discussions were added in the Results section in the revised manuscript (Please see Line 2-8, Page 6).

Response Table 1 and Supplementary Table 1. Comparison of thermal sensitivity of RENPs with other previously reported rare-earth-based luminescent nanothermometers.

Nanothermometers	Thermal sensitivity (K^{-1})	Reference
NaYbF ₄ :Er@NaLuF ₄	0.0108	This work
CaF ₂ :Yb,Er	0.0160	ACS Nano 2011 , 5 , 8665
NaLuF ₄ :Yb,Er@NaLuF ₄	0.0100	Nat. Commun. 2016 , 7 , 1047
Y ₂ O ₃ :Yb,Ho,Zn	0.0100	Dalton. Trans. 2013 , 42 , 11005
LaF ₃ :Nd	0.0026	Adv. Funct. Mater. 2015 , 25 , 615
CaF ₂ :Yb,Tm	0.0020	ACS Nano 2011 , 5 , 8665

REVIEWERS' COMMENTS:

Reviewer #2 (Remarks to the Author):

The authors address all the requests and so the recommendation of this referee is to accept the manuscript for publication as it is.

Response to the Reviewers' Reports Manuscript (NCOMMS-19-19045B)

Reviewer #2

General comments: *The authors address all the requests and so the recommendation of this referee is to accept the manuscript for publication as it is.*

Response: We appreciate the reviewer for the positive opinions on our revised manuscript.